# Selective cross-linking of coinciding protein assemblies by in-gel cross-linking mass spectrometry

Johannes F Hevler[1,2,†] ![ORCID], Marie V Lukassen[1,2,†] ![ORCID], Alfredo Cabrera-Orefice[3] ![ORCID], Susanne Arnold[3],
Matti F Pronker[1,2] ![ORCID], Vojtech Franc[1,2] ![ORCID] & Albert J R Heck[1,2,*] ![ORCID]

## Abstract

Cross-linking mass spectrometry has developed into an important method to study protein structures and interactions. The in-solution cross-linking workflows involve time and sample consuming steps and do not provide sensible solutions for differentiating cross-links obtained from co-occurring protein oligomers, complexes, or conformers. Here we developed a cross-linking workflow combining blue native PAGE with in-gel cross-linking mass spectrometry (IGX-MS). This workflow circumvents steps, such as buffer exchange and cross-linker concentration optimization. Additionally, IGX-MS enables the parallel analysis of co-occurring protein complexes using only small amounts of sample. Another benefit of IGX-MS, demonstrated by experiments on GroEL and purified bovine heart mitochondria, is the substantial reduction of undesired over-length cross-links compared to in-solution cross-linking. We next used IGX-MS to investigate the complement components C5, C6, and their hetero-dimeric C5b6 complex. The obtained cross-links were used to generate a refined structural model of the complement component C6, resembling C6 in its inactivated state. This finding shows that IGX-MS can provide new insights into the initial stages of the terminal complement pathway.

**Keywords** BN-PAGE; cross-linking; protein complexes; complement; protein modeling

**Subject Categories** Immunology; Proteomics; Structural Biology

The EMBO Journal (2021) 40: e106174

## Introduction

Over the last decades, bimolecular mass spectrometry (MS), with its ability to analyze low amounts of samples with high speed and sensitivity, has evolved into a central pillar beneficial for integrative structural biology (Lossl *et al*, 2016; Kaur *et al*, 2019; Robinson, 2019; de Souza & Picotti, 2020). The structural MS toolbox contains multiple complementary approaches. Next to native MS and top-down MS, a variety of peptide-centric MS methods, such as thermal proteome profiling (TPP), limited proteolysis (LiP), hydrogen/deuterium exchange (HDX) MS and chemical cross-linking MS (XL-MS or CLMS), have emerged and enabled structural studies of a wide range of biomolecules (Heck, 2008; Leitner *et al*, 2010; Feng *et al*, 2014; Savitski *et al*, 2014; Zheng *et al*, 2019). With recent advances in instrumentation, sample preparation, and data analysis, especially XL-MS has started to fulfill its potential to complement well-established structural methods such as X-ray crystallography, nuclear magnetic resonance spectroscopy (NMR), and cryo-electron microscopy (cryo-EM; Leitner *et al*, 2016; Matthew Allen Bullock *et al*, 2016; Rappsilber, 2011). XL-MS has a particular utility to capture protein–protein interactions in solution by measuring spatial distance restraints, mirroring structural conformations of intact proteins. Concomitantly, a wide range of chemical cross-linkers have been explored so far, often relying on similar chemical principles (Sinz, 2003; Steigenberger *et al*, 2020). Most used cross-linkers are small, homo-bifunctional reagents, with two reactive moieties capable of covalently binding two nearby amino acids. The reactive groups are separated by a spacer arm of varying lengths, which can be gas-phase cleavable or non-cleavable, thereby determining different MS data acquisition methods (Staros, 1982; Leitner *et al*, 2010; Muller *et al*, 2010; Kao *et al*, 2011). Recent advances in search engines for more efficient identification of cross-linked peptides allowed structural studies of purified proteins or protein complexes, as well as large-scale experiments with more complex samples like purified organelles or cell lysates, using buffer systems which aim to meet physiological relevant conditions (Klykov *et al*, 2018; Chen & Rappsilber, 2019; Gotze *et al*, 2019; Beveridge *et al*, 2020). A typical XL-MS workflow begins with the optimization of the cross-linker concentration. Next, a protein mixture is incubated with the

1  Biomolecular Mass Spectrometry and Proteomics, Bijvoet Center for Biomolecular Research and Utrecht Institute for Pharmaceutical Sciences, University of Utrecht, Utrecht, The Netherlands
2  Netherlands Proteomics Center, Utrecht, The Netherlands
3  Radboud Institute for Molecular Life Sciences, Radboud University Medical Center, Nijmegen, The Netherlands
   *Corresponding author. Tel: +31 302536797; E-mail: a.j.r.heck@uu.nl
   †These authors contributed equally to this work

cross-linking reagent, and the reaction is subsequently quenched to prevent the generation of unwanted random protein contacts. After (tryptic) digestion, cross-linked peptides are subjected to various pre-fractionation steps or enrichment strategies to distinguish them from the vast majority of unmodified peptides. Cross-linked residues are eventually identified using dedicated XL-MS search algorithms, providing structural information in the form of distance restraints, which can be utilized to guide computational homology modeling, refinement of flexible regions within structural models, protein–protein docking, and the generation of protein interaction networks (Bullock *et al*, 2018; Kim *et al*, 2018; Iacobucci *et al*, 2019; Albanese *et al*, 2020; Ryl *et al*, 2020). Currently, technological developments in XL-MS aim to further improve the cross-linking reaction efficiency and detection. The research is mainly focused on sample preparation techniques, MS fragmentation and enrichment strategies, data acquisition and analysis of cross-linked peptides, as well as the design of novel cross-linkers (Leitner *et al*, 2012; Liu *et al*, 2017; Iacobucci *et al*, 2018; Chen *et al*, 2019; Dau *et al*, 2019; Mendes *et al*, 2019; Steigenberger *et al*, 2019).

Although the latest advances significantly revised and reformed the field of XL-MS, some challenges remain. A central problem of XL-MS data analysis is the occurrence of both false-positive and false-negative cross-link identifications. Especially, the existence of proteins with highly dynamic/flexible conformations and the presence of co-occurring alike protein complexes (e.g., protein oligomers and co-occurring complexes sharing distinct subunits) significantly complicate the analysis by current in-solution XL-MS approaches. Interaction-specific cross-links are relevant as structural changes can be triggered by the presence of a binding partner or the environment, thereby eventually displaying a physiological relevant protein conformation (Uversky, 2011; Feng *et al*, 2014; Mannige, 2014; de Souza & Picotti, 2020). Additionally, when using too high cross-linker concentrations or too high protein concentrations, undesired artificial interactions are likely being picked up by XL-MS. In-solution XL-MS experiments, therefore, need careful experimental optimization of, in particular, the concentration of the proteins and the cross-linker. Unfortunately, these steps require

considerable sample amounts (tens of micrograms) and extra experimental time.

Here we describe an alternative approach, performing in-gel cross-linking mass spectrometry (IGX-MS), re-discovering the great separation power of gel electrophoresis. Prior to cross-linking, we load the samples and perform blue native polyacrylamide gel electrophoresis (BN-PAGE), allowing the separation of distinct structural states of the proteins or protein complexes. The distinct bands are subsequently excised and cross-linked in the gel, enabling the measurement of conformation- and interaction-specific cross-links and derived distance restraints (Fig 1). We show that this IGX-MS workflow has certain advantages compared to the in-solution based XL-MS methods. These include, among other things, no need for cross-linker concentration optimization, generation of conformation-specific cross-links, and relatively low sample amount requirements. Moreover, through IGX-MS data obtained for several protein assemblies, we provide evidence that proteins retain not only their quaternary but also their secondary and tertiary native structural states under BN-PAGE separation. To directly compare IGX-MS with in-solution XL-MS, we selected, as a proof-of-concept, the 14-mer *Escherichia coli* GroEL chaperone and different respiratory chain complexes (supercomplex 1, complex I, and complex V) from solubilized bovine heart mitochondria (BHM). Our experiments demonstrated that IGX-MS can accurately target specific subcomplexes co-existing in the protein complex mixtures and substantially reduce the number of (potentially false) over-length cross-links. Ultimately, we applied the optimized IGX-MS workflow to investigate structures of the terminal complement proteins C5 and C6, which are involved in the initial steps toward the assembly of membrane attack complex (MAC). Our cross-linking data lead us to propose a refined alternative structural conformation of the complement component C6, providing new insights into the terminal complement pathway. In summary, our data show that BN-PAGE-based IGX-MS is a powerful tool, allowing the efficient generation of compositional- and interaction-specific distance constraints, with the potential of refining structural models of a large variety of protein assemblies, even when they co-occur in solution.

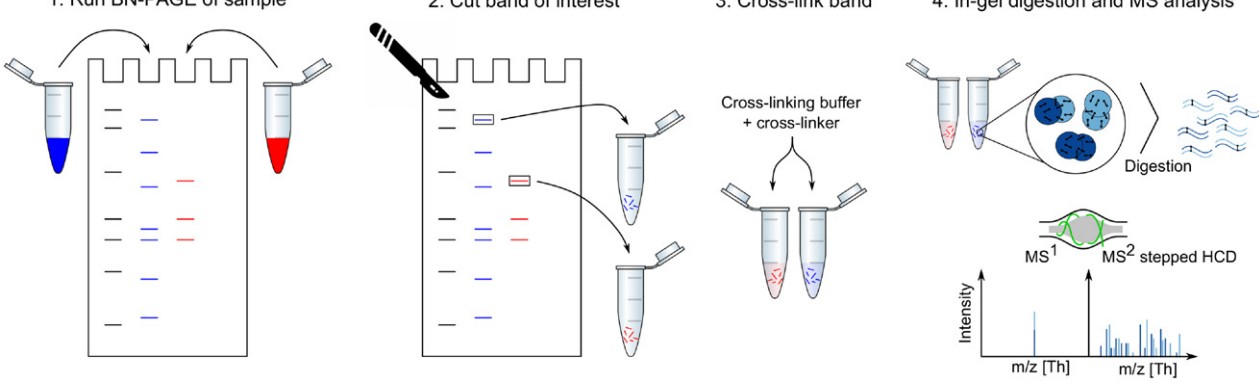

**Figure 1.  Combining BN-PAGE with IGX-MS.**
In the first step, proteins and their assemblies are loaded and separated by BN-PAGE. The bands representing different protein assemblies are visualized after running the gel with Coomassie in the upper running buffer. The band(s) of interest are then excised from the gel and incubated with a cross-linking reagent in the cross-linking buffer. The cross-linking reaction is quenched and subsequently subjected to standard in-gel digestion. The extracted peptides are finally analyzed using cross-linking optimized parameters for the MS analysis.

# Results

## BN-PAGE forms the basis for IGX-MS

Blue native polyacrylamide gel electrophoresis, BN-PAGE, has proven to be a robust and sensitive method for separating protein complexes from various sample types. It requires only minimal sample amounts to sensitively estimate native protein molecular weights (Mw), respective compositional states, and protein–protein interactions. Further, proteins and protein complexes are thought to maintain not only their overall quaternary structural organization in the gel but also their secondary and tertiary structural organization, as they can still, after that, be subjected to further structural and functional analysis (2D crystallization, cryo-EM, in-gel activity assay; Poetsch *et al*, 2000; Schafer *et al*, 2006; Wittig *et al*, 2007).

Here, we combine BN-PAGE with XL-MS to efficiently isolate and investigate co-occurring protein oligomeric states and subcomplexes. We first determined whether proteins and protein complexes can be cross-linked in a BN gel. For this, 10 μg of purified *E. coli* GroEL diluted in a Tris buffer was subjected to BN-PAGE as described previously (Wittig *et al*, 2006). Bands corresponding to the native 14-mer GroEL ($M_W$ = 800 kDa) were excised from the BN-PAGE (Appendix Fig S1A) and further cut into small pieces and incubated with or without the cross-linker reagent DSS. Next, GroEL was extracted from the gel pieces and subsequently loaded onto a reducing SDS–PAGE (Appendix Fig S1B). The control lane (no cross-linker) revealed only one distinct band at 57 kDa representing the GroEL monomers. In contrast, the BN-PAGE band that was incubated with DSS showed several additional high-molecular-weight bands above 100 kDa, indicating the successful cross-linking of GroEL subunits. Notably, the initial sample buffer (Tris) is incompatible with amine-reactive cross-linkers, as it contains primary amines that compete with the primary amines of Lysine residues, thereby significantly reducing the cross-link efficiency. Successful cross-linking of GroEL in-gel highlights the buffer exchange capacity of the BN-PAGE system, making additional buffer exchange steps (e.g., using molecular weight cut-off filters or dialysis), which would be required for in-solution cross-linking and eventually result in loss of sample, obsolete.

## IGX-MS optimization is straightforward

To prevent protein precipitation caused by over cross-linking, standard in-solution XL-MS crucially depends on using the optimal cross-linker concentration. For this, a subset of sample needs to be incubated with varying cross-linker concentrations prior to the experiment and subsequently analyzed by gel electrophoresis. The time and sample consuming optimization step led us to investigate the effect of varying cross-linker concentrations for IGX-MS experiments. GroEL was subjected to BN-PAGE, and relevant bands were cross-linked using the two different cross-linking reagent DSS and DSSO varying in five concentration steps from 0.5 to 5 mM. After quenching the cross-linking reaction with Tris, the protein-containing bands were prepared for MS analysis following a standard in-gel digestion procedure. Cross-links obtained for DSS and DSSO, and each concentration were validated by mapping them onto the GroEL structure (PDB ID: 1KP8), and lysine $C_\alpha$–$C_\alpha$ distances were obtained

(Dataset EV1). The distance distribution for both cross-linkers was highly similar at all used concentrations, and almost no cross-link distances over 30 Å were observed across the varying concentrations, indicating that IGX-MS is highly resistant against over cross-linking of proteins (Fig 2A). Our data suggest that IGX-MS is also less hampered by unspecific cross-links. We also observed that the total number of unique cross-links was not affected by the concentration of DSS, and only marginally effected for DSSO as the obtained cross-links are slightly lower for concentrations below 2 mM (Fig 2A). However, no significant difference was detected between 2 mM and 5 mM (Fig 2A). The cross-linked sites onto the GroEL structure showed good consistency in cross-linked regions for both DSS and DSSO experiments (Fig 2B, Dataset EV1). Finally, we compared the cross-linking results for each cross-linker and concentration across the three replicates, demonstrating excellent reproducibility of the IGX-MS experiments (Appendix Fig S2). Based on these results, a DSS concentration of 1.5 mM and a DSSO concentration of 2 mM (Fig 2A) were used for the subsequent experiments.

## Direct comparison of IGX-MS and in-solution XL-MS

BN-PAGE facilitates the distinction of oligomeric states, but it can also be particularly useful when protein complexes are reconstituted. In such experiments, one or more of the subunits may be (unwillingly) in excess. These preparations can then lead to false cross-link interpretations, as especially intra-cross-links can originate from the free monomer subunit or subunit in the complex (which may exhibit another conformation). By comparing IGX-MS and in-solution XL-MS of GroEL, we aimed to access the relevance of this additional separation aspect and confirm that proteins maintain their native structural integrity during in-gel separation. First, for in-solution cross-linking, GroEL diluted in Tris buffer was buffer exchanged to PBS, and subsequently, the optimal cross-linker concentration was determined by SDS–PAGE to avoid over cross-linking (Appendix Fig S3A). Next, the in-solution XL-MS sample was cross-linked with DSS, while the IGX-MS sample was cross-linked with both DSS and DSSO. Subsequent comparison of DSS- and DSSO-in-gel cross-linked samples showed a 53% overlap of identified cross-linked sites. Next, we directly compared in-solution XL-MS and IGX-MS. Although a significant number (60%) of DSS-in-gel cross-links were also detected by in-solution XL-MS (Fig 3A), in-solution XL-MS resulted in a seemingly higher total number of unique cross-links compared to IGX-MS. First, we ruled out that the higher number of unique cross-links in-solution could be explained by insufficient extraction of long peptides from the gel, based on the observation that the detected cross-linked peptides displayed a similar length distribution (Appendix Fig S3B). Plotting all the cross-links onto the GroEL structure revealed that a large portion of the "exclusive" in-solution cross-links originated from paired lysine residues separated by more than 30 Å, our distance cut-off. In contrast, virtually all IGX-MS cross-links remained below this cut-off (Fig 3B, Dataset EV1). Therefore, we are convinced that the BN-PAGE gel separation removes the co-analysis of co-occurring protein assembly states and higher-order protein aggregates. The latter often leads to the observation of over-length cross-links in solution. Further, mapping and directly comparing the overlapped cross-links between the IGX-MS and in-solution XL-MS revealed high

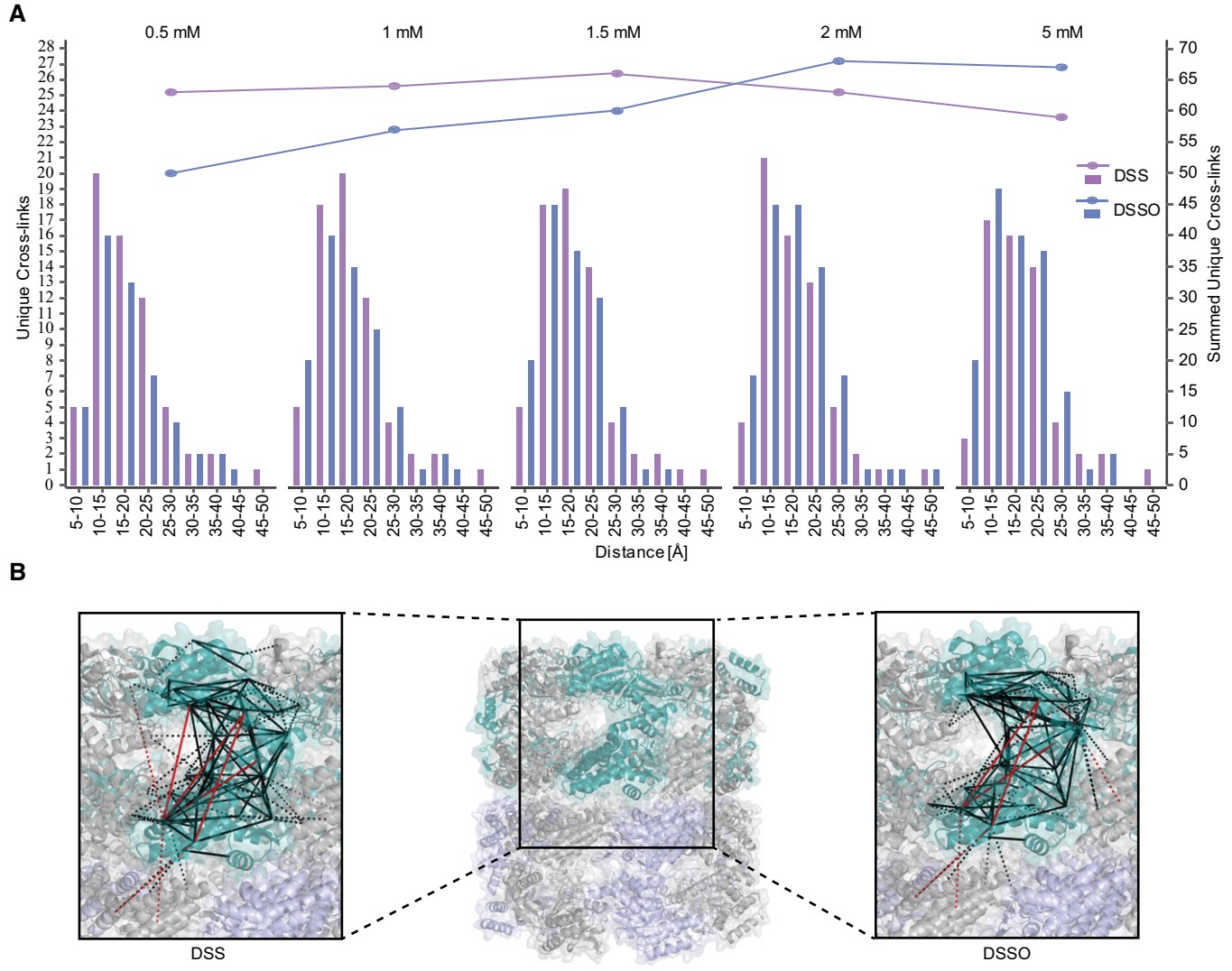

**Figure 2. IGX-MS of GroEL using either DSS or DSSO at varying concentrations.**

A *Escherichia coli* GroEL was cross-linked by IGX-MS using either DSS or DSSO at concentrations ranging from 0.5 to 5 mM. The identified cross-links were placed on the reported GroEL structure (PDB ID:1KP8), whereby the number of unique cross-linked lysine $C_\alpha$-$C_\alpha$ distances (using the left y-axis) was binned and as shown in bars. The summed number of unique cross-links obtained at each concentration is shown in lines using the right y-axis.

B Cross-links obtained by IGX-MS using DSS or DSSO plotted onto one subunit of GroEL (intra-links; solid lines) (PDB ID: 1KP8) and neighboring subunits (inter-links; dashed lines). Cross-links agreeing with the set distance restraint of 30 Å are colored black, and links exceeding the restraint are colored red.

Data information: The presented data are summed from triplicates.

consistency of cross-linked sites, supporting that GroEL preserved its native conformation in the gel.

## IGX-MS facilitates the analysis of distinct co-occurring assemblies

Proteins in cells or extracted from various biological sources can be part of multiple different complexes. Whether a protein is a free monomer or part of one or more protein complexes can substantially affect its structure. The identification of distinctive structural states of a protein by XL-MS in solution is often hampered, especially when the "free" monomer co-exists with the same protein being part of one or more complexes. For this reason, we investigated whether IGX-MS can exclusively obtain cross-links for proteins in a single configuration of the complex. As a first test sample, we incubated GroEL with one of its known natural unfolded substrates, namely the bacteriophage T4 capsid protein (gp23, 56 kDa). Following incubation of GroEL with unfolded gp23, we analyzed this sample by BN-PAGE and observed three distinctive bands, corresponding to free GroEL and GroEL with one or two copies of gp23 bound (Appendix Fig S4A). That GroEL can bind two substrate molecules (in the *cis* and *trans* ring) agrees with previously reported data (van Duijn *et al*, 2006) and could be additionally confirmed by relative quantification of

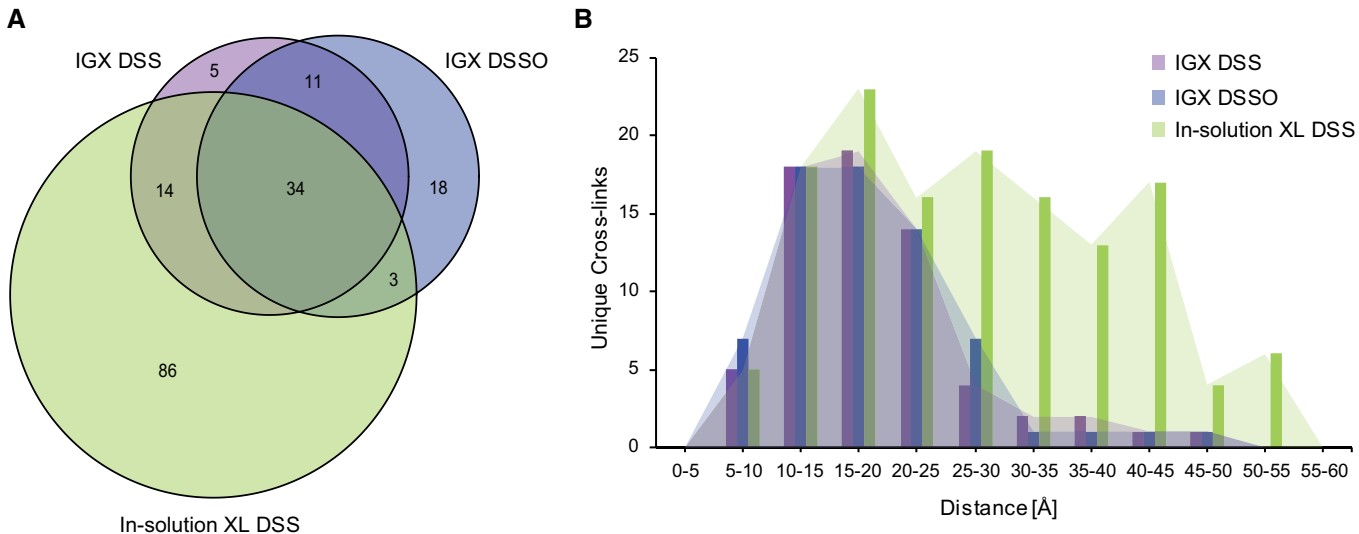

**Figure 3.  Comparison of IGX-MS and in-solution XL-MS of GroEL.**

A   *Escherichia coli* GroEL was subjected to in-solution XL-MS using 0.75 mM DSS. The resulting cross-links were compared to cross-links obtained by IGX-MS using DSS (1.5 mM) or DSSO (2 mM). The Venn diagram shows the overlap in the cross-links identified.

B   Distribution of lysine $C_\alpha$-$C_\alpha$ distances of unique cross-links identified by IGX-MS (DSS or DSSO) or in-solution XL-MS plotted on the GroEL structure (PDB ID: 1KP8).

Data information: The presented data are summed from triplicates.

the subunits in the respective bands (Appendix Fig S4B). In parallel, cross-linking of each band, i.e., GroEL, GroEL:gp23, and GroEL:(gp23)$_2$, with DSS, revealed inter-links between GroEL and gp23 exclusively in the middle and upper band whereby the primary inter-linked residues in GroEL were identified as K42, K122, and K272 (Dataset EV2). The site with most cross-links to gp23 was K272, located at the outer edge of the cavity (Appendix Fig S4C and D). IGX-MS of each BN-PAGE bands enabled the identification of protein compositional specific distance restraints, which would have been impossible by in-solution XL-MS without additional experimental steps.

Next, we assessed the capabilities of IGX on a more complex sample and subjected 20 μg of purified bovine heart mitochondria (BHM), solubilized with digitonin, to BN-PAGE (Fig 4A). It is well known that BN-PAGE can separate and visualize the different complexes of the mitochondrial respiratory chain, including many of the co-occurring supercomplexes (Schagger & Pfeiffer, 2000). The band corresponding to the monomeric form of complex V (the well-studied ATP synthase, which can also be abundantly present in a V$_2$ dimeric form), was excised and subjected to IGX-MS using DSS (Fig 4A). The detected cross-links were plotted onto the 3D structure (PDB ID: 5ARA) and compared to cross-links detected in a previously published data set from our laboratory, by in-solution XL-MS (Liu *et al*, 2018) (Fig 4B, Dataset EV3). Visualizing the detected IGX-MS- and in-solution XL-MS cross-linked regions revealed their high consistency, indicating that these membrane protein complexes largely retain their quaternary, tertiary, and secondary structures in the BN-PAGE gel. Similar to the previous in-solution XL-MS experiments, only solvent-accessible regions of complex V subunits (which in intact mitochondria are facing the matrix) were detected in the IGX data. Further, we

found that ATP5IF, a known inhibitor of the ATPase, was associated with the monomeric ATP synthase (Fig 4C). Detected cross-links from the inhibitor to ATP5F1E, ATP5F1D, ATP5F1C, and ATP5F1B agree with the previously reported binding interface of ATP5IF and the ATPase (Gledhill *et al*, 2007). In-solution XL-MS resulted in a higher number of unique cross-links (248 vs. 53 for IGX). However, like for GroEL, the $C_\alpha$-$C_\alpha$ distance distribution revealed that many in-solution XL-MS cross-links are well above the 30 Å cut-off (149 unique cross-links). The IGX-MS cross-links are predominantly below this set cut-off (only two unique cross-links above), highlighting the accordance of the IGX-MS generated restrains with the previously published structure of monomeric ATPase (Fig 4D, Dataset EV3). We argue that some of these over-length cross-links detected by in-solution XL-MS may originate from co-occurring dimeric complex V or other ATPase conformations induced upon binding of one (or several) of its many previously identified interactors (Schweppe *et al*, 2017; Liu *et al*, 2018; Ryl *et al*, 2020). Additionally, IGX-MS generated cross-links exclusively describe the interaction of ATP5IF to monomeric ATP synthase. In contrast, in-solution XL-MS generated cross-links most likely reflect distance restraints for different assembly states of ATP synthase (e.g., monomeric/dimeric ATP synthase with and without ATP5IF). To further showcase the ability of IGX-MS to generate assembly state-specific cross-links, bands corresponding to monomeric complex I (CI) and the supercomplex 1 (S1) were excised and subjected to IGX-MS using DSS (Figure EV1A). For both assembly states, a similar interaction network between the CI subunits was observed (Figure EV1A, Dataset EV3), revealing subunits that are in close proximity to each other in assembled CI (PDB ID: 5GUP). Cross-links detected for the supercomplex 1 (S1) revealed inter-complex links between NDUFB4 (a subunit of

complex I), UQCRC1, UQCRB (both subunits of complex III), COX7A1, and COX5B (both subunits of complex IV). These data agree with the previously published structure (Wu *et al*, 2016), which identified respective subunits in the interface regions of S1 (Figure EV1A, PDB ID: 5GUP). As reported for the ATP synthase, IGX-MS data for CI (monomeric and S1) closely resemble the previously reported in-solution XL-MS data (Liu *et al*, 2018; Dataset EV3). Moreover, the targeted approach of IGX-MS allowed

us to distinguish cross-links coming from monomeric CI or CI as part of S1, whereas in-solution XL-MS data represent a mixture of all the different assembly states (e.g., monomeric and S0-S4) (Fig 4A, Figure EV1B).

In summary, comparing the IGX-MS and in-solution XL-MS cross-links for mitochondrial complex V, complex I and S1 highlights the capability of IGX-MS to generate sufficient, reliable, and assembly-specific distance restraints.

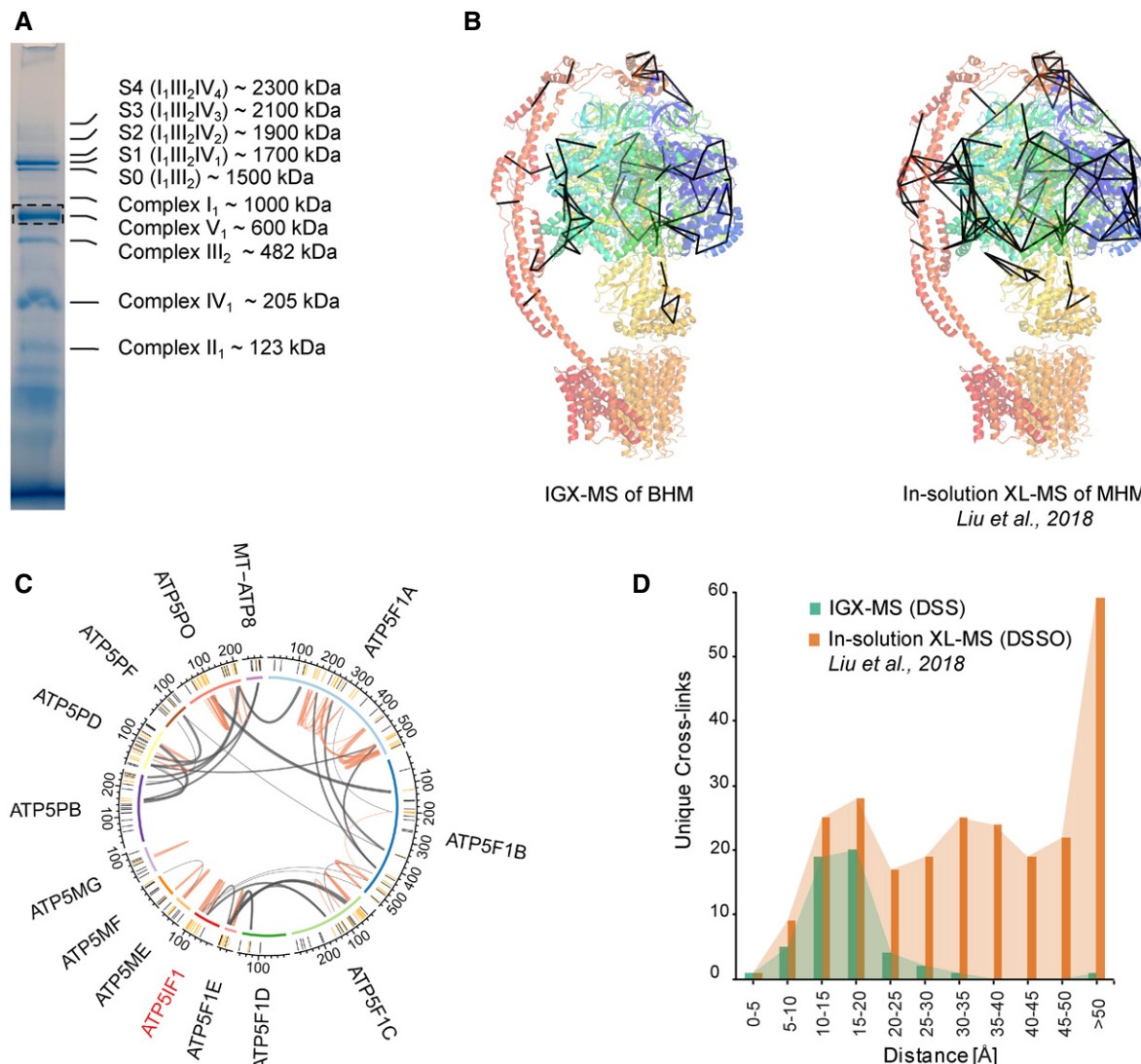

**Figure 4. Selective cross-linking of the ATP synthase monomer from bovine heart mitochondria (BHM) by IGX-MS.**

A  BN-PAGE of 20 µg solubilized BHM with several of the distinct complexes annotated. The dashed box indicates the monomeric ATP synthase (complex V) band subjected to IGX-MS. Annotation of protein bands was done following a previously published study (Wittig *et al*, 2010).

B  Unique cross-links < 30 Å identified by IGX-MS in this study (left) or by Liu *et al* (2018) using in-solution XL-MS plotted on the ATP synthase structure (PDB ID:5ARA).

C  Circos plot of cross-linked ATP synthase subunits and the ATP synthase inhibitor (ATP5IF1) identified by IGX-MS. The position of the lysine residues is shown in the outer-ring, and cross-linked residues are colored dark orange. Orange lines represent intra-links, while inter-links are colored dark gray. Thickness of the cross-link lines correlates to the number of detected cross-linked spectra matches (CSMs).

D  Distribution of lysine $C_\alpha$-$C_\alpha$ distances of unique cross-links identified by IGX-MS or in-solution XL-MS (Liu *et al*, 2018) using DSS (1.5 mM) or DSSO (0.5 mM), respectively.

Data information: The presented IGX-MS data are summed from triplicates. For illustration, the same BN-PAGE of solubilized BHM is repeatedly shown (see Fig EV1A).

## Structural features of the complement proteins C5 and C6 and how these adapt when complexed into C5b6

The terminal pathway of the complement system is mediated by sequentially interacting proteins (a.o. C5 to C9) that undergo various conformational changes in response to interactions with each other and the membrane environment (Hadders *et al*, 2012; Bajic *et al*, 2015; Schatz-Jakobsen *et al*, 2016; Bayly-Jones *et al*, 2017). In this process, membrane attack complex (MAC) is typically formed on the membrane of bacteria or pathogens, which leads to their elimination. Briefly, C6 binds to C5b, which originates from C5 by proteolytic cleavage. The resulting C5b6 complex binds C7, C8, and C9 sequentially, forming the C5b-9 complex. This complex is assembled onto the bacterial membrane and combines with polymerizing C9 molecules to create a lytic pore termed MAC (Esser, 1994). Several structures of the different components of the terminal pathway have been explored by X-ray crystallography and electron microscopy (EM; DiScipio *et al*, 1988; DiScipio & Hugli, 1989; Fredslund *et al*, 2008; Lovelace *et al*, 2011; Aleshin *et al*, 2012; Hadders

*et al*, 2012; Menny *et al*, 2018). Especially well-resolved structures of monomeric C5, C6, and C5b6 contributed to understanding conformational changes that these proteins undergo in forming the C5b6 complex (Fredslund *et al*, 2008; Aleshin *et al*, 2012; Hadders *et al*, 2012). We set out to investigate these different conformations by applying IGX-MS on monomeric C5, monomeric C6, and the hetero-dimeric C5b6 complex. Therefore, the BN-PAGE bands representing monomeric C5, monomeric C6, and C5b6 were subjected to IGX-MS (Appendix Fig S5A). Cross-links obtained for both C5 and complexed C5b were in good agreement with the respective available structural models (PDB ID: 3CU7 and 4A5W) (Fig 5A and B, Dataset EV4). In the C-terminal region of free C5, we detected some cross-links that exceed the distance restraint. This region is known to undergo significant structural rearrangements, as it adopts a more open conformation after conversion to C5b (Fig 5A, Appendix Fig S5B). Likewise, cross-links obtained for monomeric C6 and complexed C6 were plotted onto previously reported structural models (PDB ID: 3T5O and 4A5W) (Fig 5C, Dataset EV4). Cross-links obtained for C6 when present in C5b6 were consistent with the

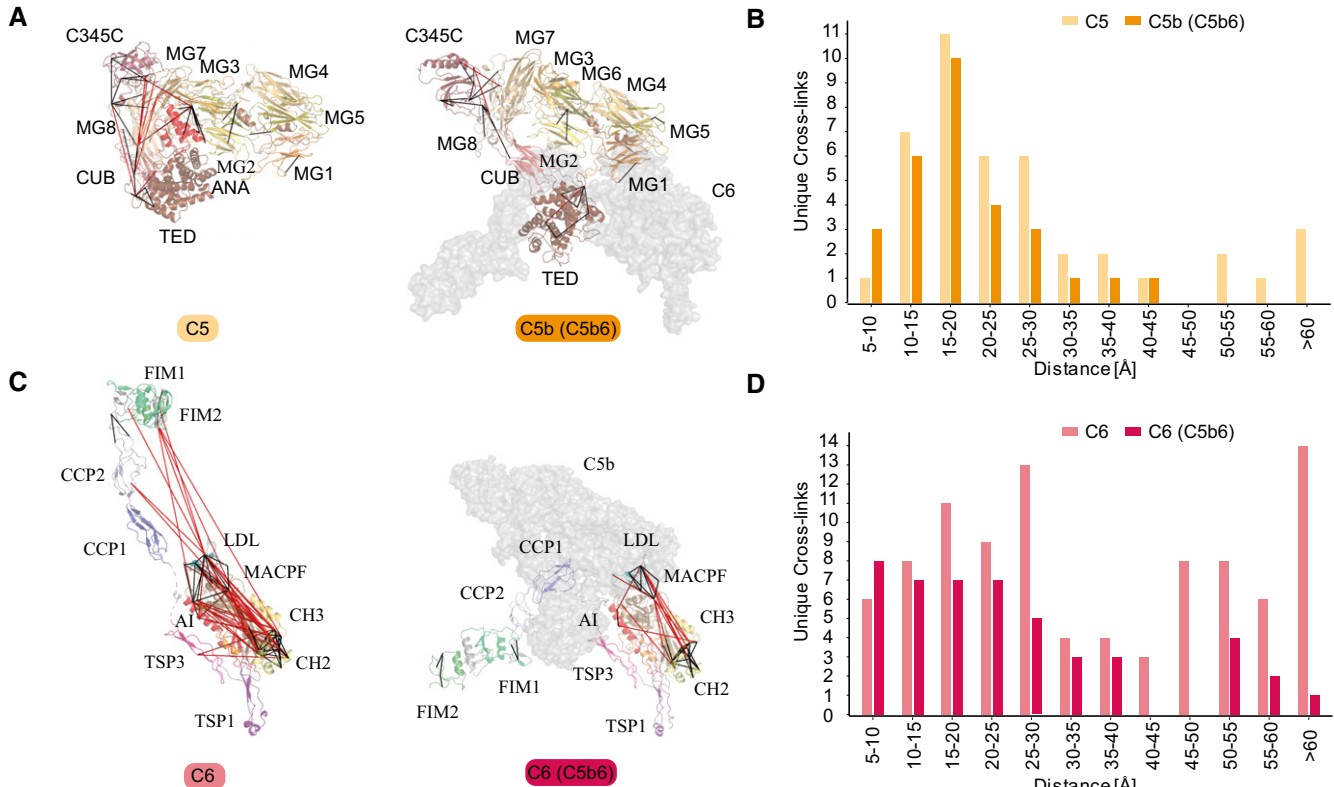

**Figure 5. IGX-MS of the monomeric complement proteins C5 and C6, and the hetero-dimeric C5b6 complex.**

A Cross-links of monomeric C5 (left) and from C5b incorporated in the C5b6 complex (right) plotted on the respective available structural models (PDB ID: 3CU7 and 4A5W, respectively). The red lines indicate distances > 30 Å. The different domains of C5 are indicated, and C6 within the C5b6 complex is shown in gray.

B Distribution of lysine $C_\alpha$-$C_\alpha$ distances of unique cross-links identified by IGX-MS in monomeric C5 (light orange) and C5b when part of the C5b6 complex (dark orange).

C Cross-links of monomeric C6 (left) and C6 incorporated in the C5b6 complex (right) plotted on the respective structural models (PDB ID: 3T5O and 4A5W, respectively). The red lines indicate distances > 30 Å. The different domains of C6 are indicated, and C5b within the C5b6 complex is shown in gray.

D Distribution of lysine $C_\alpha$–$C_\alpha$ distances of unique cross-links identified by IGX-MS in monomeric C6 (light red) or C6 when incorporated within the C5b6 complex.

Data information: Only cross-links identified in at least two out of three replicates were included in the analysis.

previously published C5b6 structure (PDB ID: 4A5W). In contrast, cross-links obtained for monomeric C6 (PDB ID: 3T5O) did not substantiate the existing structural model and showed a noticeable bimodal distance distribution (Fig 5D). Full-length C6 is composed of three thrombospondin (TSP) domains, a membrane attack complex/perforin (MACPF) domain, an LDL-receptor class A (LDL) domain, and an epidermal growth factor-like (EGF) domain. The EGF domain is followed by the C5b-binding domain composed of two complement control protein domains (CCP1 and CCP2) and two C-terminal factor I modules (FIM1 and FIM2), which are connected to the main body through a partially unresolved flexible linker (Fig 5C). Interestingly, a high number of over-length cross-links (> 30 Å) in monomeric C6 were observed between the LDL domain and the MACPF, as well as within the MACPF domain itself. Cross-links exceeding the distance constraint within the MACPF are

connecting the previously described autoinhibitory region (AI, residue 480-522) to two ∼50-residue helical clusters (CH1, residue 236–288; CH2 residue 363-416) (Fig 5C, Appendix Fig S5C) (Hadders *et al*, 2012). Secondly, several over-length cross-links were observed between the FIM2 domain and the LDL and MACPF domain residues. These over-length cross-links are nearly exclusively detected for monomeric C6 and not for C6 when present in C5b6 (Fig 5C, Appendix Fig S5C).

### Cross-link guided structural refinement of monomeric free C6

Monomeric C6 displayed an intolerably high number of over-length cross-links suggesting that an alternative conformation of monomeric free C6 may (co-)exist. Based on identifying the lysine residues involved in these over-length cross-links, such an alternative

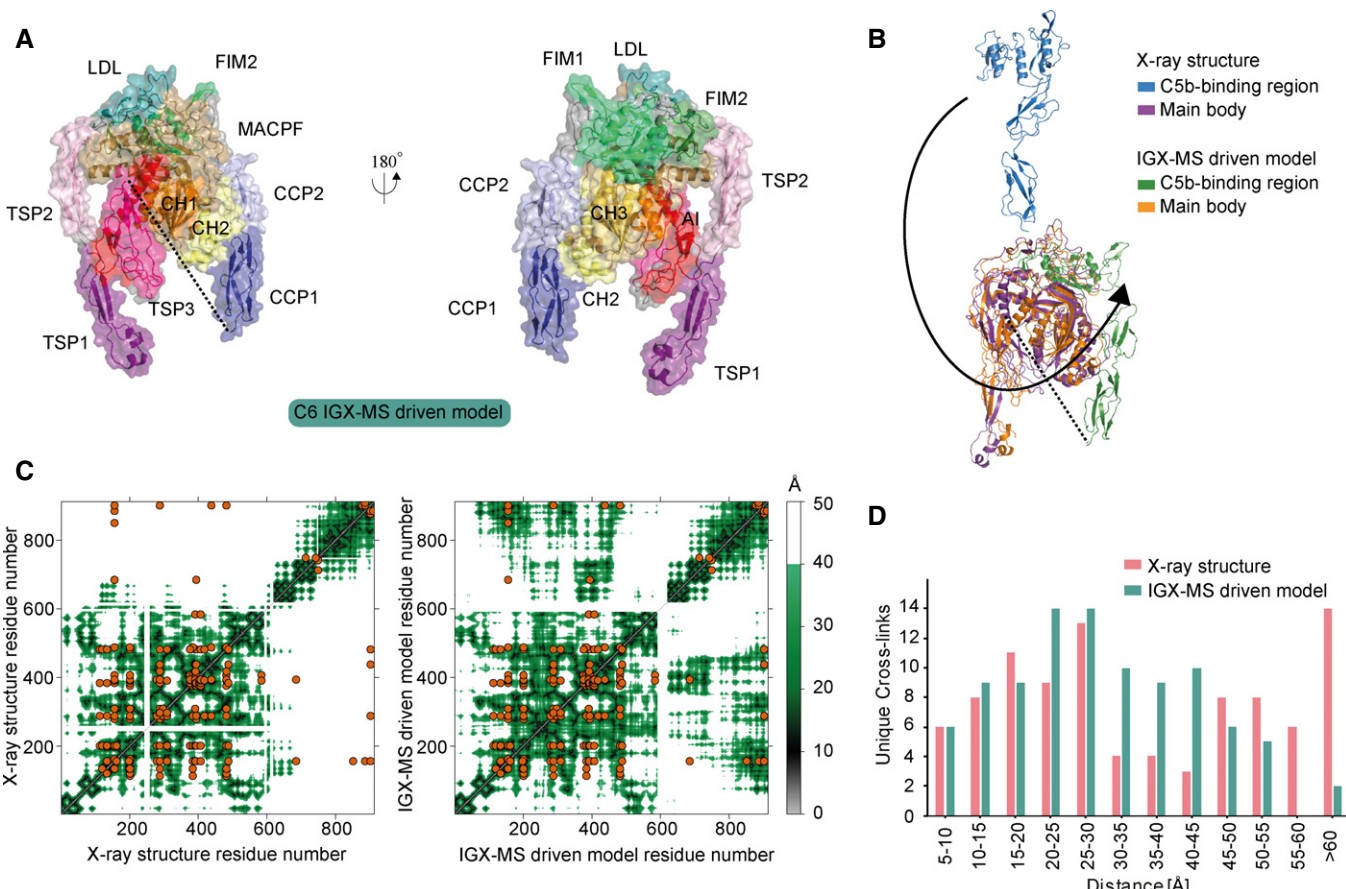

**Figure 6. IGX-MS-driven refined structural model of monomeric C6.**

A IGX-MS-driven homology model of monomeric free C6 depicted in two different orientations. Black dots indicate the few missing amino acids (residue 591–619, spanning about 81 Å) covering the linker region between the main body and C5b-binding region.

B Superpositioning of C6 IGX-MS-driven model (purple and green surface) and C6 X-ray structure (orange and blue surface; PDB ID: 3T5O). Black dots indicate the few missing amino acids (residue 591–619) covering the linker region between the main body and C5b-binding region.

C Contact maps with cross-linked residues (orange dots) of C6 X-ray structure (PDB ID: 3T5O, left panel) and IGX-MS-driven C6 model (right panel). The colored density represents a contact relationship smaller than 40 Å of individual residues. White density represents a contact relationship bigger than 40 Å of individual residues.

D Distribution of lysine Cα–Cα distances of unique cross-links identified by IGX-MS for monomeric C6 when plotted on the reported X-ray structure (pink bars, PDB ID: 3T5O) and the IGX-MS-driven refined structural model (green bars).

Data information: Only cross-links identified in at least two out of three replicates were included in the analysis.

structure would include re-positioning of the MACPF-, LDL- and the C5b-binding domains. We sought to define a structural model for monomeric C6 in two consecutive modeling steps. Our final refined model (Fig 6A) retained the characteristic sequence-specific secondary structure elements and is only missing the flexible linker domain (residues 591-619), for which no confident distance restraints were available, likely due to the lack of lysine residues in this region. The missing linker comprises 28 amino acids, resulting in an 81 Å-gap in the IGX-MS-driven model of C6 (Fig 6A—black dots). Considering an average residue length of 3.4 – 4 Å, the length of this linker translates to 95.2–112 Å, which is sufficient to accommodate the produced gap (Ainavarapu *et al*, 2007). When comparing the inter-domain rotation angles and domain centroid displacements between our model and the monomeric C6 X-ray structure (PDB ID: 3T5O), it becomes apparent that the main body (TSP1-1-TSP1-2-LDL-MACPF-EGF-TSP1-3) and the C5b-binding region (CCP1-CCP2-FIM1-FIM2) undergo a significant motion to each other (angle of 32° and displacement of 110.0 Å between TSP1-3 and FIM1—Fig 6B, Table EV1). On the other hand, the C5b-binding region moves almost like a single body with small inter-domain angles and displacements (largest angle of 2° and largest displacement of 1.1 Å). Within the main body, substantial domain reorientations can also be observed; in particular between EGF and TSP1-3 (angle of 83° and displacement of 16.7 Å), between TSP1-1 and TSP1-2 (angle of 42° and displacement of 19.7 Å), and between EGF and TSP1-3 (angle of 29° and displacement of 16.3 Å) (Fig 6B, Table EV1). Further, when comparing the MACPF domain of the IGX-MS-driven model to the previously reported structures of monomeric, complexed, and activated C6, noticeable intra-domain differences are obtained (Fig EV2A–D). Overall, the MACPF domain comprises a central four-stranded β-sheet, an AI region dominated by a linchpin helix, and three helical clusters (CH1-3) of which CH1 and CH2 unfold upon C6 activation (Fig EV2A). Closer examining the MACPF domain of the monomeric C6 X-ray structure (Fig EV2B) revealed a remarkable conformational resemblance with the MACPF domains of complexed C6 (Fig EV2C) and activated C6 (Fig EV2D). Interestingly, in our cross-linked-driven structural model, we obtained a different conformational orientation of the regions within the MACPF domain (Fig EV2A). A clear re-positioning of the linchpin helix (part of the AI region), as well as the CH1 and CH2 cluster, can be delineated when compared to the MACPF of the X-ray structure (Fig EV2E). Here, the linchpin helix of the IGX-MS-driven model (sand-colored structure) is tilted toward the central β-sheets and the helical clusters (CH1-3) (Fig EV2F). Additionally, the CH1 and CH2 domains are shown to be re-located, with the CH1 domain moved upwards, and the CH2 domain tilted toward the linchpin helix when compared to the MACPF domains of the previously reported monomeric, complexed and activated C6 structures (Fig EV2F). Conclusively, the structural rearrangements, guided by our IGX-MS data, result in a more closed conformation of the MACPF domain for free monomeric C6. Besides over-length cross-links within the MACPF domain, we detected eight cross-link restraints between the C5b-binding domain (specifically CCP2, FIM1, and FIM2) and LDL and the MACPF domain (specifically CH2; Appendix Fig S6A). The domains that are usually involved in the binding interface of C6 and C5b (CCP1 and CCP2—see Fig 5C) are in our model predicted to wrap around the MACPF domain, sharing an interaction interface with its CH2 and CH3 cluster

(Fig 6A, Appendix Fig S6B). The FIM2 domain formes an interaction interface with residues of the TSP2, LDL, and MACPF domain, thereby locking the C5b-binding domain to the main body of C6 (Fig 6A and B, Appendix Fig S6C, Dataset EV5). This observation is in sharp contrast to the reported X-ray structure, in which the C5b-binding domain shows an "elongated" conformation, with no interaction interface between the mentioned domains (Fig 6B). Further, we generated contact maps to assess the overlap of cross-link data with the C6 X-ray structure and the IGX-MS-driven model. The IGX-MS-driven model provides new contact possibilities between the C5b-binding domain and the LDL and MACPF domain as well as within the MACPF domain (Fig 6C), thereby significantly improving the overlap between the IGX and reported structural data (Fig 6D, Dataset EV4 and Dataset EV6). To validate the structural model for C6, we additionally performed an in-solution XL-MS experiment on purified C6. Firstly, the optimal cross-linker concentration was determined by incubating C6 with varying DSS concentrations (0–1.5 mM) (Fig EV3A). The optimization also revealed the formation of low amounts of dimeric C6 at all used cross-linking concentrations (Fig EV3A). Thus, an additional SDS–PAGE was performed following the in-solution cross-linking reaction (using 0.25 mM DSS) to selectively detect cross-links for monomeric C6. The in-solution generated cross-links were in agreement with the IGX-MS generated cross-links for monomeric C6, thereby suggesting a re-positioning of the MACPF-, LDL- and the C5b-binding domains (Fig EV3B). Finally, like for IGX-MS generated cross-links, the by in-solution XL-MS generated distance restraints for monomeric C6 are better satisfied for our cross-linked refined structural model than the X-ray structure (mean cross-link distance 26.2 Å vs. 41.7 Å—Fig EV3C).

## Discussion

In-solution XL-MS has become a useful tool to study protein structures and protein–protein interactions (Liu *et al*, 2014; Henry *et al*, 2018; Koukos & Bonvin, 2020). Even though the technology of in-solution XL-MS advanced substantially over the last decade, there are still quite a few challenges left. The current in-solution XL-MS method is time and labor demanding. Further, in-solution XL-MS requires a considerable amount of sample, which is necessary for optimizing the cross-linker concentration and the enrichment of cross-linked peptides. Also, in-solution co-occurring protein oligomers and distinct protein complexes within a sample can complicate the structural analysis. Unfortunately, current in-solution XL-MS workflows need further experimental steps to separate these protein complexes and potentially demand higher starting material amounts.

Here we introduce IGX-MS, an alternative approach that aims to tackle some of the challenges of in-solution XL-MS mentioned above. Firstly, we demonstrate the feasibility of cross-linking proteins in a BN-PAGE gel environment. We show that the efficiency of the in-gel cross-linking reaction is cross-linker- and also concentration-independent (DSS and DSSO work equally well). The latter makes the time- and sample consuming cross-linker concentration optimization obsolete. BN-PAGE combined with IGX-MS is very sensitive, requiring only a few micrograms of a protein sample to dissect and study different co-occurring protein complexes. These

features reduce sample preparation time as no purification of a protein/ protein complex of interest is needed and enables the analysis of samples that are only available in minimal quantities (and therefore not susceptible to conventionally used purification methods for in-solution XL-MS). Another step, which can be omitted by using IGX-MS, is the sample's buffer exchange into a cross-link compatible buffer since it occurs already in the gel. The IGX-MS data presented here are hallmarked by high reproducibility, and to a large extent, the observed cross-links agree with those found by parallel in-solution cross-links. We find this as an important conclusion. It supports that proteins in the BN-PAGE gel maintain their native quaternary, tertiary, and secondary structures, as previously suggested (Poetsch *et al*, 2000; Schafer *et al*, 2006; Wittig *et al*, 2007). Compared to in-solution XL-MS, IGX-MS shows a desired significant reduction of over-length cross-links when plotted on reported structures. A closer investigation of in-solution and in-gel cross-links revealed that IGX-MS generates protein state-specific cross-links with fewer undesired cross-linking products than in-solution XL-MS. This specificity is mainly achieved due to the option to precisely cross-link a specific oligomeric state or individual protein complex of interest. Compared to in-solution XL-MS, IGX-MS also has some caveats. Not all proteins and proteins assemblies enter a BN-PAGE gel easily, and some do not migrate as well-defined bands. Further, the resolving power of commercially available gels can provide a challenge in resolving distinct protein assemblies close in size. However, gels can be cast in-house, using gradients optimized for specific mass regions, potentially allowing a fast and easy adaption and modification suitable for specific sample/protein types.

Following proof-of-concept studies on samples, ranging from GroEL to mitochondrial respiratory chain complexes, we ultimately demonstrate that the distance restraints generated by IGX-MS can also be used to guide computational homology modeling and refine structural models. The reported structure of monomeric free C6 obtained by X-ray crystallography has been suggested to be in a partially activated and extended state (Aleshin *et al*, 2012). This partial activation was attributed to crystal lattice contacts with neighboring C6 molecules mimicking C5b in the C5b6 hetero-dimer (Appendix Fig S7). Our IGX-MS-driven refined structural model suggests a more compact structural arrangement for monomeric C6. The linchpin helix of the AI region becomes tilted with an imaginary rotational center around one of the previously identified hinge regions (Aleshin *et al*, 2012) and more closely located to the central β-sheet as well as the helical clusters (CH1, CH2), which are known to unfold in the activated MACPF complex (Menny *et al*, 2018). This arrangement indicates that the linchpin helix interacts with the CH1 domain as previously suggested (Aleshin *et al*, 2012) and also with the central four-stranded β-sheet and the CH2 domain. A closer position of the linchpin helix might hint at an autoinhibitory function, to tightly control the unfolding of respective regions. Next, the C5b-binding domain can be found wrapped around the LDL and MACPF domain, stabilizing the more compacted MACPF domain. Satisfyingly, the here presented IGX-MS-driven structural model agrees quite well with the overall shape of C6 as reported in early negative stain EM data (DiScipio & Hugli, 1989). We compared our data to previously published structures of complexed and activated C6 to obtain new insights into the dynamic activation process of C6. Upon binding of C6 to C5b, the C5b-binding domain is no longer wrapped around the LDL and MACPF domain of C6, offering a binding interface for C5b. Simultaneously, the linchpin helix rotates around the hinge region to the left, and the CH1 domain moves downwards, bringing the EGF domain of the AI and the CH1 cluster in close proximity. Further, the CH2 cluster opens up by bending slightly to the right, resulting in a more open conformation of the MACPF domain, which culminates in the complete unfolding of CH1 and CH2 to elongated β-sheets in MAC (Fig EV2A–E, Movie EV1). This initial unfolding of C6 provides an alternative conformational step in the terminal complement pathway, eventually pointing toward a new structural state of the C6 domains that is characteristic for its unbound state. Further, it would be interesting to investigate whether such a mechanism could also be true for C7, which shares a similar domain structure. Subsequently, we performed XL-MS of C6 in-solution. The observed cross-links were in very good agreement with the IGX-MS derived data, and thus also fitted well on our refined C6 structural model. The agreement between the BN-PAGE and in-solution data shows that proteins with a large conformational space can retain their native structures in BN-PAGE (Fig EV3A–C). The in-solution XL-MS of C6 also confirmed some of the caveats of in-solution XL-MS we mentioned above, as in this case DSS concentration optimization revealed formation of small amounts of C6 dimer at all used DSS concentrations (Fig EV3A). Thus, to ensure that only cross-links coming from monomeric C6 were analyzed, we implemented another gel-based experimental step (SDS–PAGE separation) for in-solution XL-MS of C6 to extract only cross-links for the C6 monomer.

The work presented here collectively describes a novel methodology termed IGX-MS, which allows the efficient, sensitive, and reproducible generation of specific structural distance restraints. The methodology described here should not be regarded as replacement for in-solution XL-MS, but rather as a convenient, alternative approach. IGX-MS can best be used on proteins and protein complexes that can be well separated in a BN-PAGE gel and this is not the case for all proteins. But when amendable to BN-PAGE, IGX-MS provides the ability to distinctively analyze co-occurring protein oligomers in purified systems, even when originating from more complex samples, such as solubilized mitochondria.

## Materials and Methods

### Materials

Chemicals and reagents were purchased from Sigma-Aldrich (Steinheim, Germany) unless otherwise stated. Acetonitrile (ACN) was purchased from Biosolve (Valkenswaard, The Netherlands). Sequencing grade trypsin was obtained from Promega (Madison, WI). NativePAGE 3–12% Bis-Tris protein gels, NativePAGE Sample Buffer, NativePAGE Running Buffer, NativePAGE Cathode Additive, and NativeMark were purchased from Invitrogen (California, USA). Criterion XT Bis-Tris Precast Gels (4–12%), XT MOPS Running Buffer, and sample buffer were purchased from Bio-Rad (California, USA). DSSO was produced in-house according to a previous protocol (Kao *et al*, 2011). Oasis HLB 96-well µElution Plates were purchased from Waters (Massachusetts, USA). GroEL and the major capsid protein gp23 were expressed and purified as previously described. (Quaite-Randall & Joachimiak, 2000; van Duijn *et al*, 2005; van Duijn *et al*, 2006) Complement components C5, C6, and

C5b6 were purchased from CompTech (Texas, USA). The fresh bovine heart was obtained from a slaughterhouse.

## Separation of proteins using Blue native PAGE (BN-PAGE)

Blue native page analysis was performed according to the manufacturer's protocol and recently published protocols (Wittig *et al*, 2006). Briefly, proteins were mixed with NativePAGE sample buffer (1x final concentration), and subsequently, 5–20 µg of protein sample was loaded onto a Bis-Tris gel (3–12%). Electrophoresis was started with a dark blue cathode buffer (1x NativePAGE cathode additive) for 30 min at 80 V before the dark blue buffer was changed to light blue cathode buffer (0.1x NativePAGE cathode additive). After the change of buffer, the voltage was increased to 120–140 V for additional 2–4 h. Readily run gels were briefly rinsed with ddH$_2$O before gel bands of interest were excised and further cut into smaller pieces under a laminar flow hood. Excised protein bands were stored in Eppendorf tubes for subsequent cross-linking experiments.

## Verification of in-gel cross-linking (IGX) by SDS–PAGE

Purified GroEL (10 µg) in Tris buffer (50 mM Tris–HCL, pH 7.7, 1 mM EDTA, 1 mM DTT, 10% glycerol) was analyzed using BN-PAGE as described above. Next, excised gel bands were incubated in 50 µl PBS with or without 1.5 mM DSS for 30 min at room temperature (RT). Then, the cross-linking reaction was quenched by addition of Tris to a final concentration of 50 mM for 30 min at RT. Next, the supernatant was removed from the gel pieces, and proteins were extracted in 200 µL extraction buffer (50 mM Tris, pH 7.9, 1 mM dithiothreitol (DTT), 150 mM NaCl, 0.1% SDS) overnight at RT. The gel pieces were separated by centrifugation at 14,000 *g* for 2 min, and the supernatant dried to 20 µL. The samples were heated to 95°C for 5 min with 50 mM DTT and sample buffer before loaded onto SDS–PAGE (4–12%). The gel was prepared according to the manufacturer's protocol using MOPS running buffer. After the SDS–PAGE separation finished, the gel was briefly washed with ddH$_2$O and subjected to Coomassie brilliant blue staining solution for approximately one hour. Stained SDS gels were de-stained in ddH$_2$O, overnight and shaking at RT.

## DSS and DSSO concentration range experiments with GroEL

BN-PAGE followed by IGX of purified GroEL was performed as described before. Excised gel pieces were incubated with an increasing concentration of DSS and DSSO (0.5, 1, 1.5, 2, and 5 mM) to determine the effect of different cross-linker concentrations. Cross-linking experiments were performed in triplicates. After quenching of cross-linking reactions, the supernatant was removed, and gel pieces were briefly washed using ddH$_2$O and subsequently subjected to standard in-gel digestion (Shevchenko *et al*, 2006). Briefly, gel pieces containing cross-linked proteins were washed, reduced by incubation in reduction buffer (50 mM ammonium bicarbonate (AmBic), 6.5 mM DTT, pH 8.5) for one hour at RT. The reduction buffer was removed, and gel pieces were dehydrated using 100% ACN. Next, dehydrated gel pieces were subjected to alkylation buffer (50 mM AmBic, 54 mM iodoacetamide (IAA), pH 8.5) for 30 min at RT in the dark. The alkylation buffer was

removed, and gel pieces were dehydrated using 100% ACN. For digestion of cross-linked proteins, dehydrated gel pieces were covered with digestion buffer (3ng/µl Trypsin in 50 mM AmBic, pH 8.5) and pre-incubated for a minimum 30 min on ice. Next, excess of digestion buffer was removed, and an equivalent volume of AmBic buffer (50 mM AmBic, pH 8.5) was added to cover the gel pieces, before incubating at 37°C overnight. Next, the supernatant containing digested peptides was collected, and gel pieces were once again dehydrated using 100% ACN for 15 min at RT. Resulted supernatant was collected and combined with the previous supernatant. Finally, the samples were completely dried and stored at −80°C until MS analysis. For MS analysis, cross-linked peptides were resuspended in MS buffer (10% FA in water) and analyzed as described below.

## LC-MS analysis

Data for IGX-MS samples were acquired using an UHPLC 1290 system (Agilent Technologies, Santa Clara, CA) coupled on-line to an Orbitrap Fusion or Orbitrap Fusion Lumos mass spectrometer (Thermo Scientific, San Jose, CA). Firstly, peptides were trapped using a 100-µm inner diameter 2-cm trap column (packed in-house with ReproSil-Pur C18-AQ, 3 µm) prior to separation on an analytical column (50 cm of length, 75 µM inner diameter; packed in-house with Poroshell 120 EC-C18, 2.7 µm). Trapping of peptides was performed for 5 min in solvent A (0.1% FA in water) at a flow rate of 0.005 ml/min. DSS cross-linked peptides were subsequently separated as follows: 0-13% solvent B (0.1% FA in 80% v/v ACN) in 10 s, 13-44% in 40 min, 44-100% in 3 min, and finally 100% for 2 min. DSSO cross-linked samples were separated using the following gradient: 0-10% solvent B in 10 s, 10–40% in 40 min, 40-100% in 3 min, and finally 100% for 2 min. For each of the gradients, the flow was passively split to approximately 200 nL/min. Mass spectrometers were operated in a data-dependent mode (DDA). For DSS cross-linked peptides, full scan MS spectra from 350-1500 Th were acquired in the Orbitrap at a resolution of 60,000 with the AGC target set to $1 \times 10^6$ and maximum injection time of 20 ms. For measurements on the Orbitrap Fusion, in-source fragmentation was turned on and set to 15 eV. Cycle time for MS$^2$ fragmentation scans was set to 3 s. Only peptides with charged states 3-8 were fragmented, and dynamic exclusion properties were set to $n = 1$, for a duration of 20 ms. Fragmentation was performed using in a stepped HCD collision energy mode (31.5, 35, 38.5%) in the ion trap and acquired in the Orbitrap at a resolution of 30,000 after accumulating a target value of $1 \times 10^5$ with an isolation window of 1.4 Th and maximum injection time of 120 ms. For the acquisition of DSSO cross-linked peptides, full scan MS spectra from 310–1,600 Th were acquired in the Orbitrap at a resolution of 120,000 with the AGC target set to $5 \times 10^5$ and maximum injection time of 50 ms. For the identification of DSSO signature peaks, peptides were fragmented using a fixed CID collision energy (30%) and MS$^2$ scan was performed at in the Orbitrap at a resolution of 30,000 after accumulating a target value of $5 \times 10^4$ ions using an isolation window of 1.6 Th and maximum injection time of 54 ms. For sequencing selected signature peaks, selected ions were fragmented using a fixed HCD collision energy (30%) in the ion trap MS$^3$, with the AGC target set to $1 \times 10^4$ and maximum injection time of 120 ms.

### In-solution XL-MS of GroEL

Purified GroEL (10μg) was cross-linked using 0-2 mM DSS for 30 min at RT, followed by quenching using a final concentration of 50 mM Tris. Cross-linked samples were analyzed by SDS–PAGE to determine an optimal cross-linker to protein ratio. The optimal DSS concentration (0.75 mM, Appendix Fig S2A) was used for cross-linking of 20 μg GroEL (1 mg/ml) in triplicates. After quenching of the reactions, protein precipitation was performed by adding three times 50 μl cold acetone and subsequent incubation at −20°C overnight. Precipitated samples were centrifuged at 12,000 $g$ for 20 min. After careful removal of the supernatant, the remaining pellet was air-dried until no acetone solution was visible anymore. Pellets were resuspended in 50 μl ABC with 0.33 μg trypsin (1:60) and incubated with shaking for 4 h at 37°C. The solubilized pellets were reduced by 5 mM TCEP for 5 min at 95°C followed by alkylation with 30 mM CAA for 30 min at 37°C. Digestion was performed overnight by 0.4 μg trypsin (1:50) at 37°C. The samples were acidified with TFA before desalting using Oasis HLB plate. Finally, the eluent was dried completely and solubilized in 10 % FA before MS analysis.

### Analysis of GroEL bound to unfolded gp23

The major capsid protein gp23 was unfolded in 8 M urea for one hour at RT. Unfolded gp23 (11.4 μM) was incubated with GroEL (0.8 μM) in Tris buffer with ADP (50 mM Tris, pH 7.5, 50 mM KCl, MgCl$_2$, 1 mM ADP) 10 min at RT. The samples were then subjected to BN-PAGE, as previously described. IGX-MS was performed on the three occurring bands, as described earlier, using 1.5 mM DSS.

### Data analysis of GroEL cross-links

Raw files obtained from IGX-MS of GroEL were analyzed with the Proteome Discoverer (PD) software suite version 2.3 (Thermo Fisher Scientific) with the incorporated XLinkX node for analysis of cross-linked peptides. For DSS data, the non-cleavable cross-link search option was used, while DSSO data were searched by the $MS^2/MS^3$ option. A FASTA file containing the GroEL sequence was used for the XlinkX search. For the samples of GroEL complexed with gp23, the FASTA file was supplemented with the sequences of GroEL and gp23. Raw files were searched with the precursor mass tolerance set to 10 ppm, the maximum FDR rate set to 1% and ΔXlinkX score ≥ 40. Carbamidomethyl was set as fixed modification and oxidation (M) and acetylation (protein N-term) as variable modifications. The obtained cross-links were plotted onto the GroEL structure (PDB ID:1KP8) to extract the Cα-Cα distances using a python script for PyMol. For further cross-link analysis, both intra-chain and inter-chain combinations to neighboring subunits were considered. In the distance histograms, only the shortest combination is represented if several possible combinations existed. When plotted on the structure, several combinations are shown if they are below 30 Å. Cross-link sequence overviews were generated using xiNET.(Combe *et al*, 2015). Data obtained from IGX-MS of GroEL bound to gp23 were searched in MaxQuant (version 1.6.10.0) to obtain iBAQ (intensity-based absolute quantification) values. Trypsin was set as a digestion enzyme with two allowed missed cleavages. Carbamidomethyl was set as fixed modification and oxidation (M) and acetylation (protein N-term)

as variable modifications. The FASTA file used for the search contained sequences of GroEL and gp23.

### Isolation and purification of bovine heart mitochondria (BHM)

The bovine heart was freshly obtained from a slaughterhouse, kept on ice for 1 h, and immediately used for mitochondria isolation. All procedures were performed within a cold room and/or maintaining the material and solutions on ice. The heart (ca. 600 g) was cut into smaller pieces while removing excess of fat and connective tissue. The pieces of cardiac muscle tissue were homogenized in 4 ml/g tissue of ice-cold isolation buffer (250 mM sucrose, 10 mM Tris/HCl pH 7.4, 0.5 mM EDTA, and 2 mM phenyl-methane-sulfonyl fluoride) using a blender at low speed for 5 s and at high speed for 1 min. The pH of the homogenate was measured and corrected to 7.4 with 2 M Tris (unadjusted). After a 15-min stirring, the homogenate was centrifuged at 400 $g$ (20 min; 4°C). The supernatants were filtered through 8 layers of gauze and centrifuged at 7,000 $g$ (30 min; 4°C). The resulting mitochondria-enriched pellets were resuspended in isolation buffer and again homogenized this time by applying 10 strokes using a Potter-Elvehjem homogenizer. The mitochondrial homogenates were centrifuged at 10,000 $g$ (20 min; 4°C), and the resulting pellets (crude mitochondria) were resuspended in isolation buffer supplemented with protease inhibitor cocktail (SIGMAFAST™). Protein concentration was determined by the DC protein assay (Bio-Rad). Aliquots were shock-frozen in liquid nitrogen and stored at −80°C. In order to increase the purity of the preparation, crude mitochondria (4 × 15 ml aliquots; ca. 60 mg prot/ml) were thawed on ice, diluted (1:4) with ice-cold washing buffer (250 mM sucrose, 20 mM Tris/HCl pH 7.4, 1 mM EDTA) and centrifuged at 1,000 $g$ (10 min; 4°C). The supernatants were recovered and centrifuged at 40,000 $g$ (20 min; 4°C), and each resulting pellet (clean mitochondria) was resuspended in 2 ml washing buffer. Afterward, mitochondria were loaded onto a two-layer sucrose gradient (1 M/1.5 M) and centrifuged at 60,000 $g$ (20 min; 4°C). The fractions accumulated at the interphase (pure mitochondria) were carefully recovered and pooled into one tube. After resuspension in 20 ml ice-cold washing buffer, pure mitochondria were centrifuged at 10,000 $g$ (20 min; 4°C) and finally resuspended in 5 ml ice-cold washing buffer supplemented with protease inhibitor cocktail (SIGMAFAST™). Protein concentration was determined as above described, and the aliquots of pure mitochondria were shock-frozen in liquid nitrogen and stored at −80°C.

### IGX-MS and data analysis of ATP synthase isolated from purified BHM

Purified bovine heart mitochondria were solubilized with digitonin (9 g/g protein) on ice for 30 min. Subsequently, 20 μg of solubilized mitochondria was analyzed using BN-PAGE. Afterward, a band corresponding to the ATP synthase was excised, and IGX-MS was applied as described above using 1.5 mM DSS. Triplicates were measured, and individual raw files (corresponding to a specific gel band) were searched in MaxQuant (Cox & Mann, 2008) against the *Bos Taurus* proteome (2019_08, downloaded from UniProt) with a PSM FDR of 1%. Trypsin was set as a digestion enzyme with two allowed missed cleavages. Carbamidomethyl was set as fixed modification and oxidation (M) and acetylation (protein N-term) as

variable modifications. Proteins identified for each raw.file were subsequently used to generate a "reduced" fasta.file for the cross-link search in PD using the XlinkX node using previously described settings. Identified cross-links corresponding to the ATP synthase, complex I and S1 complex (I-III$_3$-IV) were subsequently extracted and plotted onto the previously published structure (PDB ID: 5ARA, 5GUP). Resulting Cα-Cα distances were compared to previously published in-solution data for cross-linked mouse heart mitochondria (Liu *et al*, 2018). An overview of cross-linked subunits for mentioned proteins was generated using the "circlize" package for R (Gu *et al*, 2014).

## IGX-MS and data analysis of complement proteins

Complement components C5 (5 μg), C6 (5 μg), and C5b6 (10 μg) were subjected to BN-PAGE followed by IGX-MS as previously described using 1.5 mM DSS. Experiments were done in triplicates. The resulting raw files from the MS analysis were searched in MaxQuant (version 1.6.10.0) to generate libraries for the C5, C6, and C5b6 bands. The data were searched against the reviewed *Homo Sapiens* UniProt database (2019_08, downloaded from UniProt). Trypsin was set as a digestion enzyme with two allowed missed cleavages. Carbamidomethyl was set as fixed modification and oxidation (M) and acetylation (protein N-term) as variable modifications. The data were then searched using PD, as described earlier. Mannosylation of tryptophan residues was added as a variable modification, and MaxQuant generated libraries used in the XLinkX search. Only cross-links observed in two out of the three replicates were included for further analysis. The cross-links were plotted onto the respective structures using PyMol to obtain Cα-Cα distances. Cross-link sequence overviews were generated using xiNET.(Combe *et al*, 2015).

## Modeling of an alternative structure of free complement C6

Cross-links derived for free C6, together with additional structural constraints derived from UniProt (disulfide-bond information, secondary structure elements; UniProt Accession: P13671), were used to predict an alternative structural model. Briefly, the modeling process was divided into two consecutive steps. First, an I-Tasser homology model of C6 based on the previously published structure (PDB ID: 3T5O) was generated to resolve missing residues (Yang *et al*, 2015). Next, a flexible linker region (residue 591–619) and the C5b-binding domain (residues 620–913) were removed from the generated C6 model. Subsequently, regions with a high density of cross-linked residues were removed from the shortened C6 structure, producing a "core-template" for comparative modeling using Modeller 9.24 (Webb & Sali, 2016). Excised regions were provided as additional templates (Table EV2) to support the modeling process together with the cross-linking restraints (mean = 17 Å, stdev = 2) obtained for individual residues (1-590) as well as the secondary structure information which was obtained from UniProt (UniProt ID: P13671). In total, 20 cross-linked guided models for free C6 were generated, each first optimized with the variable target function method (VTFM) and afterward refined using molecular dynamics (MD) optimization (Sali & Blundell, 1993). For each model, a DOPE score and a GA341 score were calculated to further validate the quality of produced models (Melo *et al*, 2002; John & Sali, 2003;

Shen & Sali, 2006). Additionally, contact maps for each one of the 20 models were generated, and a CM score (Schweppe *et al*, 2016) was calculated, indicating the overlap of the cross-linking data with the respective contact maps using the XLmap package in R (Schweppe *et al*, 2016) (Dataset EV7). The model satisfying both scores the best (DOPE and CM score) was chosen for the second modeling process, to generate a full-length model of C6 using detected cross-links between C6 (residues 1–590) and the C5b-binding domain (residues 620–913). The structural assembly of both was achieved by predicting an interaction interface by DisVis (van Zundert & Bonvin, 2015) using respective cross-links and solvent-accessible residues as input parameters. Solvent-accessible residues were identified using the standalone program Naccess (© S. Hubbard and J. Thornton 1992-6). Residues with relative solvent accessibility ≥ 40% were used as solvent-accessible residues. Finally, information-driven docking with HADDOCK (Karaca & Bonvin, 2011; van Zundert *et al*, 2016) with the validated cross-links and the identified active residues was performed, resulting in four distinct clusters. A file (.json) containing all parameters set for the protein docking was deposited to the ProteomeXchange partner PRIDE database (for details, see "Data availability" section). The structure showing the best agreement with the distance restraints used for the docking process and the best Haddock score was chosen as final model (Dataset EV8). Additionally, residues participating in a binding interface between the C6 (residues 1-590) and the C5b-binding domain were predicted using the Prodigy webserver (Dataset EV5). For final model validation, all IGX-MS derived cross-links obtained for C6 were plotted onto the model, and distances for respective links were compared to the X-ray structure of C6 (PDB ID 3T5O) (Dataset EV6).

## Characterization of inter-domain rotation angles

From the comparison of our IGX-MS-driven model with the crystal structure of C6 in isolation (PDB 3T5O), inter-domain rotation angles and centroid displacements were determined by sequentially superposing the domains (with indicated boundaries) of the crystal structure of C6 onto the corresponding domains of our model, using the program Superpose (Krissinel & Henrick, 2004), part of the CCP4 suite (Winn *et al*, 2011). Superposition was based on C$_\alpha$ atoms of indicated domains (Table EV1). To further compare the protein conformations, a distance map was generated using the Bio3D package for R (Grant *et al*, 2006). For this, coordinates of superimposed MACPF domains (residue 155-501) extracted from the crystal structure, and our model were provided as input (Figure EV2E).

## In-Solution XL-MS of C6

Purified C6 (5 μg, 0.33 mg/ml) was cross-linked using 0–1.5 mM DSS for 30 min at RT, followed by quenching using a final concentration of 50 mM Tris. Cross-linked samples were analyzed by SDS–PAGE to determine an optimal cross-linker to protein ratio. Lower DSS concentrations down to 10 μM were also tested. The optimal DSS concentration (0.25 mM, Fig EV3A) was used for cross-linking of 20 μg C6 (0.33 mg/ml) in triplicates. After quenching the reactions, 5 μg was separated by SDS–PAGE, and the monomeric band was subjected to in-gel digest before XL-MS/MS analysis. Raw files were analyzed as previously described for IGX-MS of complement

proteins using MaxQuant and the XlinkX node of PD using previously described settings. Identified cross-links were plotted onto the previously published crystal structure (Dataset EV4) and our IGX-MS-driven model (Dataset EV6).

## Data availability

The mass spectrometry data from this publication have been deposited to the ProteomeXchange partner PRIDE database (Vizcaino et al, 2016) and assigned to the identifier PXD020014 (https://www.ebi.ac.uk/pride/archive/projects/PXD020014).

**Expanded View** for this article is available online.

## Acknowledgments
All authors acknowledge support from the Netherlands Organization for Scientific Research (NWO) funding the Netherlands Proteomics Centre through the X-omics Road Map program (project 184.034.019) and the EU Horizon 2020 program INFRAIA project Epic-XS (Project 823839). MVL thanks Independent Research Fund Denmark (project 9036-00007B).

## Author contributions
JFH and AJRH conceptualized the study. JFH and MVL designed the methodology, performed experiments, and analyzed the data. AC-O and SA provided the mitochondrial samples. MFP performed the characterization of inter-domain rotation angles. JFH and MVL wrote the original draft. JFH, MVL, MFB, AC-O, SA, VF, and AJRH carefully revised and edited the manuscript before submission. MVL and AJRH acquired funding and resources. AJRH supervised the project.

## Conflict of interest
The authors declare that they have no conflict of interest.

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
