## [Review Process File · The EMBO Journal]

Selective cross-linking of coinciding protein assemblies by in-gel cross-linking mass-spectrometry

Johannes Hevler, Marie Lukassen, Alfredo Cabrera-Orefice, Susanne Arnold, Matti Pronker, Vojtech Franc, and Albert Heck

DOI: [10.15252/embj.2020106174](https://doi.org/10.15252/embj.2020106174)

Corresponding author(s): Albert Heck (a.j.r.heck@uu.nl)

Review Timeline:	Submission Date:	7th Jul 20
	Editorial Decision:	17th Aug 20
	Revision Received:	17th Oct 20
	Correspondence:	23rd Nov 20
	Correspondence:	25th Nov 20
	Editorial Decision:	27th Nov 20
	Revision Received:	3rd Dec 20
	Accepted:	10th Dec 20

Editor: Hartmut Vodermaier

Transaction Report:

Thank you again for submitting your manuscript on in-gel cross-linking mass spectrometry to The EMBO Journal. Three referees with biological/structural (#1) and proteomics (2&3) have now provided the comments copied below, in light of which we would be interested in pursuing this study further for our Resource section, pending adequate addressing of a number of specific points raised in the reports. As you will see, referees 1 and 2 have mostly more minor concerns, but referee 3 remains at this stage still unconvinced of the superiority of the in-gel method over conventional in-solution crosslinking-MS approaches; and while I am less worried regarding his/her skepticism about more or less straightforward workflow/sample preparation, I feel it would be important to add at least one experiment comparing the methods side-by-side and proving that IGX-MS can generate biological insights that classical in-solution XL-MS could not, e.g. along the lines suggested in referee 3's last paragraph.

REFEREE REPORTS

Referee #1:

This manuscript by Hevler et al. presents a new strategy for structural mass spectrometry that combines blue-native PAGE with IGX-MS. They use two model systems (GroEL and ATP synthase) to demonstrate that tertiary structure of proteins remains intact during this method and confirm their observed crosslinks with in solution XL-MS. The main advantage of their method lies in harnessing the high-resolution separation of Blue-native PAGE to select a single species for cross-linking MS analysis. This method will be a powerful approach to mapping interaction interfaces on complex assemblies especially when used in conjunction with cryoEM maps of moderate or near-

atomic resolution.

The authors go on to apply their method to investigate the conformation of complement proteins. Specifically, they analyse crosslinks of monomeric C5, monomeric C6, and cross-links present in the C5b6 complex. In doing so they discover an alternative conformation of C6, which is described as an auto-inhibited structure. This differs from the published C6 crystal structure, predominantly in the arrangement of the auxiliary domains (CCP1/2 and FIM1/2) which wrap around the core MACPF of C6 in their model. They also describe an alternative conformation of the linchpin helix of C6 and of the helical bundles which unfurl in the final MAC. Overall this study presents an exciting new finding that reveals a new conformation of soluble C6 that could provide insight into how the early steps in MAC assembly are regulated. I have a few comments/suggestions that would improve the manuscript further:

Results:

The new alternative conformation of C6 requires a significant rearrangement of the CCP and FIM domains of C6. These domains are connected to the core MACPF by a flexible linker of 28 residues, which are not modelled here due to lack of cross-linked lysines. It would help the non-expert reader if you could put a sentence or so describing how the length of the linker could accommodate the big distance that needs to be covered here. For example, a scale bar in Figure 6 together with a statement about the length of the peptide backbone for a linker of that length if it was fully extended.

Figure 6A: The legend states that 3 different orientations of C6 are shown, but your figure shows only 2.

SI Figure 1: The inset (-) and (+) above the gel in B should be specifically mentioned in the legend.

SI Figure 6: A-D the colouring here does not match the legend. The CH2 is not dark pink in any of these panels (it is yellow, green, light green and can't make out the colour in D). CH3 is not green. You have it as grey in A-D. The MACPF domain looks tan not pink. I also found panel F very difficult to follow your colouring. It might improve readability if all of the domains in panels A-D had the same colour scheme and in F you chose a different colour for each conformation. Then with the black labels, you should be able to distinguish the areas.

SI Figure 7C: What are highlighted as sticks in this panel? Are these the side chains of the cross-linked lysines? I found it difficult to understand how to relate the boxed areas with the regions indicated in the structure.

SI Figure 8: The legend says it is highlighting the interaction of the CCP and FIM domains, but these are not specifically indicated on the structure. The only colour coding for C6 is N to C rainbow. Please label these domains on the image to aid the non-expert reader.

Discussion section:

The authors discuss the crystal structure of C6 and rationalize that it could be a partially active form due to lattice contacts. It would also be worth mentioning that the buffer conditions of the crystal structure included high concentrations of divalent cations (CdCl₂). Another divalent cation Mg is particularly important in convertase activation. Divalent cations may serve a broader purpose and also prime the "opening" of C6 to facilitate binding the newly cleaved C5b. The rapid

association of C6 is essential for stabilizing C5b (Cooper & Muller-Eberhard, J Exp Med 1970).
<https://www.ncbi.nlm.nih.gov/pmc/articles/PMC2138854/>

Also, the authors should discuss their structural model for C6 within the context of early negative stain information for fluid-phase C6 (Discipio & Hugli JBC 1989).
<https://www.jbc.org/content/264/27/16197.long>

The structural model presented here agrees very well with the compact shape observed in these earlier negative stain images.

Referee #2:

Summary

Heck and co-workers introduce a new concept for chemical cross-linking (in combination with mass spectrometry): in-gel cross-linking. Protein complexes are separated by native PAGE, bands of interest are cut out, and the cross-linking step is performed in the gel. Following in-gel proteolysis, cross-linked peptides are identified in the same way as for the conventional in-solution cross-linking workflow. A side-by-side comparison of in-solution and in-gel cross-linking using the GroEL complex as an example reveals important findings, including (1) the optimization of cross-linking conditions in the gel is relatively straightforward as the numbers of identified cross-links do not change much for different reagent concentrations; (2) overall, cross-link data suggests that the conformation of protein complexes is preserved in the native gel; (3) cross-links tend to show fewer "outliers" compared to in-solution cross-linking that arise from compositionally and/or conformationally heterogeneous populations.

The initial findings from the GroEL experiments form the basis for further applications of the new method. First, the authors show that structurally valid cross-links are obtained from small amounts of input material for mitochondrial supercomplexes. Second, the authors perform an in-depth analysis of a binary complex of members of the complement system (C5b-C6). The in-gel cross-linking experiments suggest that a much larger conformational heterogeneity exists than what is reflected in published 3D structures obtained by X-ray crystallography and electron microscopy. Distance restraints derived from the cross-linking experiments are then used for modeling the structure of C6 protein.

Overall, the present work is clearly original and addresses an important topic in cross-linking mass spectrometry - the fact that in-solution cross-linking of heterogeneous samples results in convoluted data that may be derived from different assembly states. Although mutually incompatible cross-link information can to some extent be distinguished at the data analysis stage, this is neither trivial nor universally applicable. Combining BN-PAGE and in-gel cross-linking provides an interesting alternative to in-solution processing, and given the increasing importance of chemical cross-linking in structural and systems biology, the method will be of interest to many.

The manuscript is well-written and provides appropriate details and references, unless mentioned in my comments below. Overall, I have mostly minor comments.

Specific comments

Major

1. Conceptually, it would be interesting to have another comparison of in-gel and in-solution cross-linking at the end of the manuscript, for the complement proteins. An in-solution XL experiment should also reveal cross-links that do not match the known structures, maybe more, maybe less. This would be an additional confirmation that the in-gel cross-links are truly derived from native conformations of proteins that have a large conformational space (in this respect, GroEL might be considered structurally more rigid I assume).

Minor

2. Page 4, last line: "sample concentration requirements" - the authors probably meant sample amounts rather than concentration?
3. Page 5: The abbreviation MAC does not make sense in this context, as it stands for membrane attack complex (explained on page 13).
4. Page 5, legend to Figure 1: rephrase "running the gel by Coomassie in the upper running buffer"
5. Page 5, results: It is not obvious that in general, information on conformational states can be derived from BN-PAGE. Compositional (e.g. oligomericity) states, yes, but conformational would imply different migration based on the conformation.
6. Page 8: In this manuscript, XL-MS is used as a synonym for in solution cross-linking (as opposed to in-gel). I find this rather confusing, as in-gel cross-linking would also fall under the commonly used meaning of XL-MS. I suggest to always use the term in solution XL-MS for clarity.
7. Page 8: Why would a Gaussian distance distribution be expected for cross-linked GroEL?
8. Page 10, last line: How is a "major site of interaction" defined?
9. Page 11: The statement in the first three lines is somewhat vague, i.e. that solution XL-MS cannot be used to generate "protein assembly specific" restraints. It is not clear what "assembly" refers to here, if a cross-linked complex could be separated after cross-linking into distinct bands (in a denaturing or non-denaturing gel), then this would also work.
10. Page 14, legend to Figure 5: In contrast to the other applications, the authors used a subset of cross-links observed in at least 2 out of 3 replicates for the complement protein complex (rather than an aggregate of 3 replicates). Was that done because of lower reproducibility?
11. Page 17, legend to Figure 6A: Only two, not three orientations are shown.
12. Page 18: I would disagree with the statement in the first paragraph that no separation method can be used to combine the separation of protein complexes with in solution XL-MS. As mentioned above, BN-PAGE may also be used to separate different products after XL.
13. Page 23: Regarding the data analysis by XLinkX, the authors refer to "standard settings" for

most parameters. At least mass tolerances and false discovery rates should be explicitly specified here.

14. Page 25: "previously described settings to generate a subtracted library ...". A reference or a more detailed explanation should be given here.

15. Page 26, first line: for free > from free

16. Page 26: The authors used a number of modeling strategies for C6. It may be impossible to provide details for all steps, but I feel a somewhat more extensive description might be justified for this section. For example, "information-driven docking with HADDOCK ... was performed". This is clearly not sufficient for anyone to reproduce the results.

17. Page 27, data deposition: The deposition of MS data in PRIDE is appreciated. I had a quick look at the dataset and suggest that the authors add a table with a clear description of the individual files, as the file names are not always self-explanatory.

18. SI Figure 3: It is not easy to compare the different types of cross-links in (B). Maybe it would be more informative as a line chart rather than a bar chart?

19. SI Figure 8: The legend ends abruptly, some text appears to be missing: "FIM doma..."

20. The SI video that is mentioned in the text does not seem to be available for review?

Referee #3:

In their manuscript "Selective cross-linking of coinciding protein assemblies by in-gel cross-linking mass-spectrometry" Hevler et al. investigate how cross-linking mass spectrometry (XL-MS) can be combined with blue native PAGE in-gel crosslinking (termed IGX-MS by the authors) in contrary to the conventionally used in-solution XL-MS.

The authors go on to demonstrate the applicability and use of IGX-MS on GroEL, ATP synthase from bovine heart mitochondria and the complement system. The authors show that IGX-MS, while overall leading yielding less cross-links that conventional in-solution XL-MS, leads to more robust results, circumvents critical steps in sample processing and, importantly, a substantial reduction in artificial or over-length cross-links.

While the idea of combining XL-MS with native PAGE separation is innovative and novel and highly intriguing, it is currently unclear if IGX-MS does indeed offer the advantages described by the authors and, more importantly, if those are indeed associated with potential novel biological insights that could not be gained from conventional XL-MS.

Generally, it appears therefore doubtful if this approach, while exciting for the more MS-oriented community, could warrant a publication in EMBO J.

Specifically,

The authors convincingly show that IGX-MS leads to very reproducible crosslinks under varying crosslinker concentrations. However, in order to state that this is an advantage over in-solution XL-MS, the authors have to repeat these experiments with in-solution XL-MS and show that this is here not the case.

Regarding the argument for a simplified work-flow. This is a bit of a peculiar argument, as IGX-MS necessitates the use of an additional PAGE gel - such an extra, arguably quite time and work-intensive step. In contrary to in-solution XL-MS, where, as the name indicates, the sample is directly cross-linked and processed in solution.

Also, the other arguments, e.g. that there is no need for an extra buffer-exchange nor for time-consuming test of crosslinker concentrations are only half-convincing - see above - and as it has been shown that crosslinking also works in all kinds of buffer environments, including TRIS. It is also not convincingly shown in the current manuscript, how IGX-MS can work with smaller amounts of sample than in-solution XL-MS - given the fact that it requires this additional gel fractionation step.

The main argument of the authors, that IGX-MS leads to a substantial reduction in artificial or over-length cross-link, is indeed a potentially significant advantage, but also needs clarification.

The authors compare their data on ATPsynthase to an earlier dataset of their own (Liu et al., 2018). This is interesting, as the authors have explicitly stated in that paper that the majority of their crosslinks (137 of 139) was below the critical threshold of 30 Å. It would be nice if the authors were to clarify this point.

For the other datasets generated in this study - e.g. on GroEL - would the amount of over-length crosslinks for in-solution XL-MS be reduced if the authors were to crosslink GroEL at lower concentration?

Or in other words - the authors have used relatively large amounts of sample at relatively high-concentrations - and according to the workflow the sample concentrations during in-solution XL-MS (prior to native PAGE) were obviously different to the ones used for IGX-MS (e.g. after native PAGE). This should be taken into account.

Still, IGX-MS, also in the eyes of the reviewer, is a truly neat idea and could hold the potential to generate biological meaningful data. However, in order to prove this, the authors should carry out an additional experiment showing that is possible to obtain data with IGX-MS that in-solution XL-MS is not able to generate.

For example, perform IGX-MS of a cell lysate and demonstrate that IGX-MS can differentiate between different populations of the same complex.

Rebuttal letter

Remarks of the reviewers in black, our responses in blue.

Referee #1:

This manuscript by Hevler et al. presents a new strategy for structural mass spectrometry that combines blue-native PAGE with IGX-MS. They use two model systems (GroEL and ATP synthase) to demonstrate that tertiary structure of proteins remains intact during this method and confirm their observed crosslinks with in solution XL-MS. The main advantage of their method lies in harnessing the high-resolution separation of Blue-native PAGE to select a single species for cross-linking MS analysis. This method will be a powerful approach to mapping interaction interfaces on complex assemblies especially when used in conjunction with cryoEM maps of moderate or near-atomic resolution.

The authors go on to apply their method to investigate the conformation of complement proteins. Specifically, they analyse crosslinks of monomeric C5, monomeric C6, and cross-links present in the C5b6 complex. In doing so they discover an alternative conformation of C6, which is described as an auto-inhibited structure. This differs from the published C6 crystal structure, predominantly in the arrangement of the auxiliary domains (CCP1/2 and FIM1/2) which wrap around the core MACPF of C6 in their model. They also describe an alternative conformation of the linchpin helix of C6 and of the helical bundles which unfurl in the final MAC. Overall this study presents an exciting new finding that reveals a new conformation of soluble C6 that could provide insight into how the early steps in MAC assembly are regulated. I have a few comments/suggestions that would improve the manuscript further:

Results:

The new alternative conformation of C6 requires a significant rearrangement of the CCP and FIM domains of C6. These domains are connected to the core MACPF by a flexible linker of 28 residues, which are not modelled here due to lack of cross-linked lysines. It would help the non-expert reader if you could put a sentence or so describing how the length of the linker could accommodate the big distance that needs to be covered here. For example, a scale bar in Figure 6 together with a statement about the length of the peptide backbone for a linker of that length if it was fully extended.

We assume that the average length of an amino acid backbone is 3.4 – 4 Å and the linker is therefore expected to be 95.2 – 112 Å. This is enough to accommodate the gap of 81 Å in the model structure. This statement with reference is added to page 10 – line 283-287. The length of the gap was also added in legend corresponding to Fig 6A.

Figure 6A: The legend states that 3 different orientations of C6 are shown, but your figure shows only 2.

We fixed this mistake. The legend was changed to “two different orientations”

SI Figure 1: The inset (-) and (+) above the gel in B should be specifically mentioned in the legend.

The legend was changed to: “The non-cross-linked control (-) showed only a band at 57 kDa of the GroEL subunit, whereas in the DSS-cross-linked sample (+) reveals several bands at higher Mw.”

SI Figure 6: A-D the colouring here does not match the legend. The CH2 is not dark pink in any of these panels (it is yellow, green, light green and can't make out the colour in D). CH3 is not green. You have it as grey in A-D. The MACPF domain looks tan not pink. I also found panel F very difficult to follow your colouring. It might improve readability if all of the domains in panels A-D had the same colour scheme and in F you chose a different colour for each conformation. Then with the black labels, you should be able to distinguish the areas.

SI Fig 6 (now Figure EV2) was changed according to the suggestions by the reviewer. The color of the structures in A-D are the same color and the different conformations in panel F have different colors. The legend was corrected accordingly.

SI Figure 7C: What are highlighted as sticks in this panel? Are these the side chains of the cross-linked lysines? I found it difficult to understand how to relate the boxed areas with the regions indicated in the structure.

The residues highlighted as sticks (now Appendix Fig S6) are the same as the residues written in the black boxes. These residues were predicted to be part of the interaction interface (C5b6-binding region to MACPF- and LDL-domain) of the C6 model. The legend was adapted to further clarify this annotation.

SI Figure 8: The legend says it is highlighting the interaction of the CCP and FIM domains, but these are not specifically indicated on the structure. The only colour coding for C6 is N to C rainbow. Please label these domains on the image to aid the non-expert reader.

We followed the reviewer's advice. The domain labels of FIM, CCP and MACPF of the rainbow-colored C6 was added to SI Figure 8 (now Appendix Fig S7).

The authors discuss the crystal structure of C6 and rationalize that it could be a partially active form due to lattice contacts. It would also be worth mentioning that the buffer conditions of the crystal structure included high concentrations of divalent cations (CdCl₂). Another divalent cation Mg is particularly important in convertase activation. Divalent cations may serve a broader purpose and also prime the “opening” of C6 to facilitate binding the newly cleaved C5b. The rapid association of C6 is essential for stabilizing C5b (Cooper & Muller-Eberhard, J Exp Med 1970).

<https://www.ncbi.nlm.nih.gov/pmc/articles/PMC2138854/>

The reviewer's reference hints at the stabilization of C5b by C6 but does not show the influence of cations on this interaction. We agree that magnesium may play a role in complement activation but cannot find support in the literature for the statement regarding divalent cation induced “opening” of C6. To avoid speculations, we keep with the argument from the original

paper on the C6 x-ray structure (Aleshin et al., 2012) with lattice contacts as the main reason for partial activation.

Also, the authors should discuss their structural model for C6 within the context of early negative stain information for fluid-phase C6 (Discipio & Hugli JBC 1989).

<https://www.jbc.org/content/264/27/16197.long>

The structural model presented here agrees very well with the compact shape observed in these earlier negative stain images.

We agree and appreciate this was pointed out. The compact C6 model does indeed agree with the overall shape of the early negative stain data. This observation and reference were added to the discussion (page 13 – line 399-401).

Referee #2:

Summary

Heck and co-workers introduce a new concept for chemical cross-linking (in combination with mass spectrometry): in-gel cross-linking. Protein complexes are separated by native PAGE, bands of interest are cut out, and the cross-linking step is performed in the gel. Following in-gel proteolysis, cross-linked peptides are identified in the same way as for the conventional in-solution cross-linking workflow. A side-by-side comparison of in-solution and in-gel cross-linking using the GroEL complex as an example reveals important findings, including (1) the optimization of cross-linking conditions in the gel is relatively straightforward as the numbers of identified cross-links do not change much for different reagent concentrations; (2) overall, cross-link data suggests that the conformation of protein complexes is preserved in the native gel; (3) cross-links tend to show fewer “outliers” compared to in-solution cross-linking that arise from compositionally and/or conformationally heterogeneous populations.

The initial findings from the GroEL experiments form the basis for further applications of the new method. First, the authors show that structurally valid cross-links are obtained from small amounts of input material for mitochondrial supercomplexes. Second, the authors perform an in-depth analysis of a binary complex of members of the complement system (C5b-C6). The in-gel cross-linking experiments suggest that a much larger conformational heterogeneity exists than what is reflected in published 3D structures obtained by X-ray crystallography and electron microscopy. Distance restraints derived from the cross-linking experiments are then used for modeling the structure of C6 protein.

Overall, the present work is clearly original and addresses an important topic in cross-linking mass spectrometry - the fact that in-solution cross-linking of heterogeneous samples results in convoluted data that may be derived from different assembly states. Although mutually incompatible cross-link information can to some extent be distinguished at the data analysis stage, this is neither trivial nor universally applicable. Combining BN-PAGE and in-gel cross-linking provides an interesting alternative to in-solution processing, and given the increasing importance of chemical cross-linking in structural and systems biology, the method will be of interest to many.

The manuscript is well-written and provides appropriate details and references, unless mentioned in my comments below. Overall, I have mostly minor comments.

We thank the reviewer for this positive evaluation.

Specific comments

Conceptually, it would be interesting to have another comparison of in-gel and in-solution cross-linking at the end of the manuscript, for the complement proteins. An in-solution XL experiment

should also reveal cross-links that do not match the known structures, maybe more, maybe less. This would be an additional confirmation that the in-gel cross-links are truly derived from native conformations of proteins that have a large conformational space (in this respect, GroEL might be considered structurally more rigid I assume).

We followed the reviewer's advice and included several new experimental data in the revised manuscript. To complement our direct comparison of the in-solution and in-gel cross-linking approach, we performed, as the reviewer suggested, an extra in-solution XL-MS analysis of C6. Somewhat surprising, while optimizing the cross-linker concentration, we detected on the SDS-PAGE gel a presence of low levels of dimeric C6 at all applied concentrations, including the lowest tested concentration (100 μ M) (Figure EV3A). Therefore, after in-solution cross-linking C6 by 0.25 mM DSS, the sample was separated by SDS-PAGE, and only the monomeric band was in-gel digested and analyzed by LC-MS/MS. This extra SDS-PAGE step was done to avoid possible detection of undesired cross-links originating from this artifactual C6 dimer. The in-solution XL-MS of C6 shows, consistently with IGX-MS, cross-links between the C5b-binding region (FIM and CCP domains) to the MACPF domain (Figure EV3B). Furthermore, the in-solution cross-links fit better with our IGX-MS driven model than the X-ray structure (Figure EV3C). This new additional data confirms that the in-gel cross-links indeed are derived from native conformations present in-solution.

The new data was incorporated into the manuscript as Figure EV3, in the C6 model result section (page 11 – line 331-342), and to the discussion section (page 14 – line 413-423). The method description was also added to the method section. The new data was also submitted to PRIDE (see updated Data accessibility section).

We also included new data on mitochondrial respiratory complexes, as described further below.

Page 4, last line: "sample concentration requirements" - the authors probably meant sample amounts rather than concentration?

The sentence was changed to "sample amount requirements".

Page 5: The abbreviation MAC does not make sense in this context, as it stands for membrane attack complex (explained on page 13).

We modified the sentence to further clarify that C5 and C6 are involved in the initial steps towards the complete assembly of the membrane attack complex (MAC). It now reads:

"Ultimately, we applied the optimized IGX-MS to investigate structures of the terminal complement proteins C5 and C6, which are involved in the initial steps towards the assembly of membrane attack complex (MAC)." – page 3 – line 74-76.

Page 5, legend to Figure 1: rephrase "running the gel by Coomassie in the upper running buffer"

The legend was changed to "running the gel with Coomassie in the upper running buffer"

Page 5, results: It is not obvious that in general, information on conformational states can be derived from BN-PAGE. Compositional (e.g. oligomericity) states, yes, but conformational would imply different migration based on the conformation.

We agree that compositional changes is a better term to be used. This was adapted in the manuscript on page 3 – line 80. However, we still feel that there could be cases where BN-PAGE separate conformational changes (e.g., extended vs. elongated).

Page 8: In this manuscript, XL-MS is used as a synonym for in solution cross-linking (as opposed to in-gel). I find this rather confusing, as in-gel cross-linking would also fall under the commonly used meaning of XL-MS. I suggest to always use the term in solution XL-MS for clarity.

For clarity, the term in-solution XL-MS is now used throughout the manuscript to describe cross-linking in solution. Now that we have several experiments comparing in solution cross-linking with in-gel cross-linking, we find it best to keep using the distinctive (in-solution) XL-MS and IGX-MS abbreviations.

Page 8: Why would a Gaussian distance distribution be expected for cross-linked GroEL?

We agree that the distribution does not have to be precisely Gaussian, but could also be more Maxwell-Boltzmann probability distribution. What we are aiming to say is that the observed length distribution should follow a continuous probability distribution. For clarification, the Gaussian statements were removed from the manuscript.

Page 10, last line: How is a "major site of interaction" defined?

A major site of interaction is, in this case, the residue with most inter-linked cross-links to the substrate. This sentence was rephrased in the manuscript as follows: "The site with most cross-links to gp23 was K272, located at the outer edge of the cavity (Appendix Fig S4C-D)." (page 6 – line 180)

Page 11: The statement in the first three lines is somewhat vague, i.e. that solution XL-MS cannot be used to generate "protein assembly specific" restraints. It is not clear what "assembly" refers to here, if a cross-linked complex could be separated after cross-linking into distinct bands (in a denaturing or non-denaturing gel), then this would also work.

We indeed feel we need to clarify this statement. We aim to highlight one of the benefits of in-gel XL-MS, namely the separation of protein compositional states without the need for additional experimental steps (as it would be the case for in-solution XL-MS)

We changed this in the text to: "IGX-MS of each BN-PAGE bands enabled the identification of protein compositional specific distance restraints, which would have been impossible by in-solution XL-MS without additional experimental steps." (page 6-7 – line 181-183)

Page 14, legend to Figure 5: In contrast to the other applications, the authors used a subset of cross-links observed in at least 2 out of 3 replicates for the complement protein complex (rather than an aggregate of 3 replicates). Was that done because of lower reproducibility?

The complement data were used for detailed structural analysis and modeling. Therefore, we decided to apply even stricter filtering of cross-links to ensure the confidence and quality of the downstream structural modeling.

Page 17, legend to Figure 6A: Only two, not three orientations are shown.

This was a mistake, well spotted. The legend was changed to “two different orientations” – (Fig 6A)

Page 18: I would disagree with the statement in the first paragraph that no separation method can be used to combine the separation of protein complexes with in solution XL-MS. As mentioned above, BN-PAGE may also be used to separate different products after XL.

We agree. This section has been rephrased to “Unfortunately, current in-solution XL-MS workflows need further experimental steps to separate these protein complexes and potentially demand higher starting material amounts.” (page 12 – line 352-354)

Page 23: Regarding the data analysis by XLinkX, the authors refer to "standard settings" for most parameters. At least mass tolerances and false discovery rates should be explicitly specified here.

We added information about the precursor mass tolerance and FDR to the “Data analysis of GroEL cross-links” method section (the same settings are used for all cross-link searches performed in the manuscript) (page 18 line 546-547)

Page 25: "previously described settings to generate a subtracted library ...". A reference or a more detailed explanation should be given here.

We added a more detailed explanation to the method section “IGX-MS and data analysis of ATP synthase isolated from purified BHM” (page 19-20 – line 593-599).

Page 26, first line: for free > from free

Our mistake. A correction was added as suggested by the reviewer

Page 26: The authors used a number of modeling strategies for C6. It may be impossible to provide details for all steps, but I feel a somewhat more extensive description might be justified for this section. For example, "information-driven docking with HADDOCK ... was performed". This is clearly not sufficient for anyone to reproduce the results.

To address this point, the used Haddock parameter file, describing all the parameters set for docking of the C6 FIM domain to the C6 MACPF domain, was also deposited to PRIDE so that other modelers should be able to repeat the experiments.

Page 27, data deposition: The deposition of MS data in PRIDE is appreciated. I had a quick look at the dataset and suggest that the authors add a table with a clear description of the individual files, as the file names are not always self-explanatory.

A file containing a table of a description of files submitted to PRIDE is now included.

SI Figure 3: It is not easy to compare the different types of cross-links in (B). Maybe it would be more informative as a line chart rather than a bar chart?

We followed the suggestion and panel B of SI Figure 3 (now Appendix Fig S3) was changed to a line chart.

SI Figure 8: The legend ends abruptly, some text appears to be missing: "FIM doma..."

The last sentence of the legend was changed to “The surface (grey) of two adjacent C6 molecules in the electron density maps shows the interaction of the C-terminal CCP and FIM domains of one C6 molecule with the MACPF domain of another C6 molecule.” (now Appendix Fig S7)

The SI video that is mentioned in the text does not seem to be available for review?

The video is now attached and a description of the video was added to the Supplemental information. (now Appendix Movie S1).

Referee #3:

In their manuscript "Selective cross-linking of coinciding protein assemblies by in-gel cross-linking mass-spectrometry" Hevler et al. investigate how cross-linking mass spectrometry (XL-MS) can be combined with blue native PAGE in-gel crosslinking (termed IGX-MS by the authors) in contrary to the conventionally used in-solution XL-MS.

The authors go on to demonstrate the applicability and use of IGX-MS on GroEL, ATP synthase from bovine heart mitochondria and the complement system. The authors show that IGX-MS, while overall leading yielding less cross-links than conventional in-solution XL-MS, leads to more robust results, circumvents critical steps in sample processing and, importantly, a substantial reduction in artificial or over-length cross-links.

While the idea of combining XL-MS with native PAGE separation is innovative and novel and highly intriguing, it is currently unclear if IGX-MS does indeed offer the advantages described by the authors and, more importantly, if those are indeed associated with potential novel biological insights that could not be gained from conventional XL-MS. Generally, it appears therefore doubtful if this approach, while exciting for the more MS-oriented community, could warrant a publication in EMBO J.

We thank the reviewer for the critical evaluation of our work. We are pleased that the reviewer pointed out the novelty of our innovative approach. Although the reviewer does not appear to be thoroughly convinced about the value of our work, we believe that the additional data and text editions of the criticized parts provide clarification of the high significance of our work for a broad Embo Journal readership. We are convinced that our approach deserves close attention from the non-MS-oriented community. With this work, we also aim at open up cross-linking MS technology specifically to biologists and biochemists who often have a minimal amount of sample, by far not sufficient enough to apply all desired structural analysis tools. Standard in-solution cross-linking MS experiments typically require hundreds to thousands of μg , which is, in many cases, the ultimate limitation. We demonstrate that IGX-MS can comfortably work with low μg amounts. Additionally, we show that IGX-MS can avoid generating undesired cross-links originating from artifactual protein assemblies or aggregates and provides more relevant specific structural information on selected protein complexes, from which new biological insights can be obtained.

Specifically

The authors convincingly show that IGX-MS leads to very reproducible crosslinks under varying crosslinker concentrations. However, in order to state that this is an advantage over in-solution XL-MS, the authors have to repeat these experiments with in-solution XL-MS and show that this is here not the case.

The concentration dependency when cross-linking a protein sample in-solution is well-accepted and has been described in the literature. Most described protocols for in solution XL-MS include a cross-linker concentration optimization step to avoid reaction conditions that lead to over-cross-linking of the sample (Iacobucci et al., 2018; Klykov et al., 2018).

Regarding the argument for a simplified work-flow. This is a bit of a peculiar argument, as IGX-MS necessitates the use of an additional PAGE gel - such an extra, arguably quite time and work-intensive step. In contrary to in-solution XL-MS, where, as the name indicates, the sample is directly cross-linked and processed in solution.

We respectfully disagree. The first step of in-solution cross-linking lies within finding an optimal cross-linker concentration (or a condition like pH or temperature – not discussed here), to check that cross-linking has occurred sufficiently but without over cross-linking the sample. Therefore, a subset of the sample is typically incubated with varying concentrations of cross-linker reagent. Cross-linked samples (and a non – cross-linked control) are subsequently analyzed by SDS-PAGE and potentially by BN-PAGE (especially useful for purified proteins). The aim is to determine a cross-linker concentration that produces cross-linked species without causing non-specific protein aggregation (usually visible as a smear on the gel or as thick band that has not entered the gel). It is essential to verify that no over cross-linking has occurred as cross-links from protein aggregates do not reflect the native protein conformations (Iacobucci et al., 2018; Klykov et al., 2018). Only after the optimal cross-linker concentration has been determined, the final in-solution cross-linking can be performed on a sufficient amount of sample. For IGX-MS, this sample and time-consuming optimization step is obsolete and concentration-independent in the tested concentration ranges. Thus, in-solution XL-MS requires at least two steps (concentration optimization and in-solution cross-linking experiment with optimal concentration), whereas IGX-MS only requires one step. We highlighted this problem in the main text with the following sentences: “Additionally, when using too high cross-linker concentrations or too high protein concentrations, undesired artificial interactions are likely being picked up by XL-MS. In-solution XL-MS experiments, therefore, need careful experimental optimization of, in particular, the concentration of the proteins and the cross-linker. Unfortunately, these steps require considerable sample amounts (tens of micrograms) and extra experimental time.” (page 2 – line 52-56)

Also, the other arguments, e.g. that there is no need for an extra buffer-exchange nor for time-consuming test of crosslinker concentrations are only half-convincing - see above - and as it has been shown that crosslinking also works in all kinds of buffer environments, including TRIS.

We agree, that depending on the cross-linker's chemical structure, different buffer systems might work (including Tris). However, for the here applied reagents (DSS and DSSO), which reaction chemistry is amongst the most commonly used cross-linkers (NHS-reactive cross-linkers) (Steigenberger et al., 2020), it is crucial to avoid primary amines in the buffer system (e.g., present in Tris buffer). Such primary amines compete with the primary amines of the Lysine side chain and quench the reactive sites of the cross-linker, thereby significantly decreasing the reaction efficiency. Hence, it is crucial to remove any primary amine-containing reagent before a cross-linking experiment, meaning that if the desired sample is buffered with, e.g., Tris, an additional buffer exchange step is necessary to remove competing amines (Iacobucci et al., 2018; Klykov et al., 2018). As shown in the manuscript, this is not needed for IGX-MS, since the BN-PAGE serves as a natural buffer-exchange system (see, for instance, the GroEL result section). We further highlighted this in the main text: “Notably, the initial sample buffer (Tris) is

incompatible with amine-reactive cross-linkers, as it contains primary amines that compete with the primary amines of Lysine residues, thereby significantly reducing the cross-link efficiency. Successful cross-linking of GroEL in-gel highlights the buffer exchange capacity of the BN-PAGE system, making additional buffer exchange steps (e.g., using molecular weight cut-off filters or dialysis), which would be required for in-solution cross-linking and eventually result in loss of sample, obsolete.” (page 4 – line 103-109).

It is also not convincingly shown in the current manuscript, how IGX-MS can work with smaller amounts of sample than in-solution XL-MS - given the fact that it requires this additional gel fractionation step.

To illustrate the sample and time requirements to perform the cross-linking experiment, we describe in this context additional experiment on C6 incorporated in the revised manuscript as an example:

For IGX-MS, a total amount of 15 µg (5 µg/ replicate) of protein was used to generate a sufficient amount of distance restraints to drive structural modeling.

For in-solution XL-MS of C6, we first optimized cross-linker concentration by incubating C6 with six different DSS concentrations (0-1.5 mM DSS), which resulted in 30 µg of C6 in total used for optimization (Figure EV3A). Next, 20 µg of C6 (in triplicates) was cross-linked with 0.25 mM of DSS in a total volume of 20 µL (ensuring comfortable sample handling and sufficient material for downstream processing such as digestion and de-salting), increasing the amount of used protein to 90 µg. Since cross-linking in-solution induced slight dimerization of C6, an additional SDS-PAGE separation was used to analyze monomeric, cross-linked C6, thereby further increasing the sample amount and experimental time.

The main argument of the authors, that IGX-MS leads to a substantial reduction in artificial or over-length cross-link, is indeed a potentially significant advantage, but also needs clarification. The authors compare their data on ATP synthase to an earlier dataset of their own (Liu et al., 2018). This is interesting, as the authors have explicitly stated in that paper that the majority of their crosslinks (137 of 139) was below the critical threshold of 30 Å. It would be nice if the authors were to clarify this point.

We would like to clarify: The reported cross-links (139) referred to by the reviewer are derived from Complex I, Complex II, Complex III, and Complex IV as well as from Succinyl-CoA Synthetase and MAC-ETF complex – here a citation from Liu et al (Liu et al., 2018): “After interrogating the mitochondrial organization, we also assessed the structural validity of the cross-links based on the residue-to-residue distance limit imposed by the DSSO cross-linker. For this purpose, we selected six stable protein complexes (Succinyl-CoA synthetase, MCAD-electron transferring flavoprotein complex, and ETC CI, CII, CIII, and CIV) and mapped the identified cross-links onto available high-resolution structures.”). Cross-link distances for the ATPase (PDB ID: 5ARA) presented in our study were not reported in the original manuscript by Liu et al., and therefore our data comparison with Liu et al. data set is relevant and correct. We specified the data set which we used from Liu et al. 2018 as follows: “The detected cross-links were plotted onto the 3D structure (PDB ID: 5ARA) and compared to cross-links detected in a

previously published data set from our lab, by in-solution XL-MS (Liu et al., 2018) (Fig. 4B, SI Table S3 – contains data from Liu et al. 2018 used for the comparison).”

For the other datasets generated in this study - e.g. on GroEL - would the amount of over-length crosslinks for in-solution XL-MS be reduced if the authors were to crosslink GroEL at lower concentration? Or in other words - the authors have used relatively large amounts of sample at relatively high-concentrations - and according to the workflow the sample concentrations during in-solution XL-MS (prior to native PAGE) were obviously different to the ones used for IGX-MS (e.g. after native PAGE). This should be taken into account.

We believe, if lower cross-linker concentrations were to be used for in-solution XL-MS the general number of identified cross-links would most likely go down (including both, cross-links satisfying- and violating the distance restraint). Also, optimization of the DSS concentration for GroEL (Appendix Fig 3A) in-solution cross-linking experiments did not indicate protein aggregation, which would subsequently result in a higher proportion of cross-links that are disagreeing with the structural model. The concentration of GroEL during in-solution XL-MS was chosen to be 1 mg/ml. This is the concentration that was also used for the optimization of the cross-link reaction (Appendix Fig 3A), producing different species of cross-linked GroEL subunits, without visible aggregate formation (Appendix Fig 3A). For IGX-MS 10 ug GroEL was loaded on the BN-PAGE. The concentration of GroEL during the cross-linking in the gel is determined by the size of the gel band, which was not measured. We can therefore not comment on the comparison on the concentrations as mentioned by the reviewer. However, we are convinced that we designed the experiment rigorously. We carefully evaluated the cross-linker concentration's influence on the successful cross-linking reaction in IGX-MS (Fig 2). Based on these results, we can conclude that IGX-MS provides a high number of relevant cross-links and substantially decreases the amount of over-length cross-links with no need to optimize the cross-linker concentration.

Still, IGX-MS, also in the eyes of the reviewer, is a truly neat idea and could hold the potential to generate biological meaningful data. However, in order to prove this, the authors should carry out an additional experiment showing that is possible to obtain data with IGX-MS that in-solution XL-MS is not able to generate. For example, perform IGX-MS of a cell lysate and demonstrate that IGX-MS can differentiate between different populations of the same complex.

To further showcase the ability of IGX-MS to distinguish between different populations of the same complex, we additionally applied IGX-MS on two assembly states of the mitochondrial complex I (Figure EV1). Like for the ATPase, CI protein populations were separated by BN-PAGE and bands corresponding to the monomeric complex I and the complex I in an assembled “super-complex” state with complex III and complex IV (S1 complex – I_1 -III₂-IV₁) were isolated and subjected to IGX-MS (Figure EV1A). IGX-MS enabled us to distinguish between cross-links that were only detected in either one of these assembly states (monomeric complex I or in S1) and cross-links that were detected for both assembly states (Figure EV1A-B). Again, IGX-MS generated data showing an agreeable overlap with previously generated in-solution data (Liu et al., 2018), with the advantage of distinguishing the protein (super) complex the cross-link was

detected in (Figure EV1B). We hope this data may help to convince the reviewer that IGX-MS has indeed, in several exciting cases, advantages over in solution XL-MS.

References

- Aleshin, A.E., Schraufstatter, I.U., Stec, B., Bankston, L.A., Liddington, R.C., and DiScipio, R.G. (2012). Structure of complement C6 suggests a mechanism for initiation and unidirectional, sequential assembly of membrane attack complex (MAC). *J Biol Chem* *287*, 10210-10222.
- Iacobucci, C., Gotze, M., Ihling, C.H., Piotrowski, C., Arlt, C., Schafer, M., Hage, C., Schmidt, R., and Sinz, A. (2018). A cross-linking/mass spectrometry workflow based on MS-cleavable cross-linkers and the MeroX software for studying protein structures and protein-protein interactions. *Nat Protoc* *13*, 2864-2889.
- Klykov, O., Steigenberger, B., Pektas, S., Fasci, D., Heck, A.J.R., and Scheltema, R.A. (2018). Efficient and robust proteome-wide approaches for cross-linking mass spectrometry. *Nat Protoc* *13*, 2964-2990.
- Liu, F., Lossel, P., Rabbitts, B.M., Balaban, R.S., and Heck, A.J.R. (2018). The interactome of intact mitochondria by cross-linking mass spectrometry provides evidence for coexisting respiratory supercomplexes. *Mol Cell Proteomics* *17*, 216-232.
- Steigenberger, B., Albanese, P., Heck, A.J.R., and Scheltema, R.A. (2020). To Cleave or Not To Cleave in XL-MS? *J Am Soc Mass Spectrom* *31*, 196-206.

Thank you again for submitting your revised manuscript EMBOJ-2020-106174R, "Selective cross-linking of coinciding protein assemblies by in-gel cross-linking mass-spectrometry", and also for your patience during its external and editorial reconsideration. As you will see from the comments below, while referee 2 is happy with the revised version, referee 3 was not fully satisfied by several responses to their original comments. Having discussed these persistent concerns of referee 3 now with referee 2, who shares similar expertise, I came to conclude that further experimental revision shall not be essential for publication of this resource; but before proceeding with (pre-)acceptance, I would nevertheless be interested in hearing your views and comments on the points listed in the report below, and suggestions how they might be further clarified in a final re-revised version (additional data may be helpful in instances where you should already have some).

I would thus appreciate a brief, informal response once you will have had a chance to consider the points copied below - ideally over the course of this week.

REFEREE REPORTS

Referee #2 (Report for Author)

This revised version of the manuscript by Hevler et al. addressed all my previous comments.

Specifically, additional experiments were performed and their results strengthen the case for the practical utility of in-gel cross-linking; additional experimental details have been provided as requested, and the text has been revised based on my suggestions.

In summary, I consider the manuscript acceptable for publication.

Referee #3 (Report for Author)

While the reviewer appreciates the comments and the additional experiments of the authors, unfortunately the main issues raised by the reviewer have not been addressed by the authors.

Importantly, the authors still do not substantiate their major claims: that IGX-MS has advantages over conventional in-solution XL-MS and leads to more robust results, circumvents critical steps in sample processing, requires less material and leads to a substantial reduction in artificial over-length cross-links.

In order to substantiate their claims, the authors need to compare the IGX-MS and in-solution XL-MS under fair and comparable terms.

While it is certainly advantageous to optimize crosslinker concentrations it is also widely accepted that a ratio of crosslinker/ protein does exist that - at least in the case of recombinant proteins or protein complexes - leads to adequate results in

nearly all of the cases tested (e.g. published) to date and usually does successfully avoid over crosslinking (1 to 2 ml/mg protein concentration at 1 to 2 mM crosslinker concentration).

Thus, the correct comparison is to compare crosslinks from in-solution XL-MS under these standard conditions with IGX-MS - and then compare which work-flow is simpler, requires less material and leads to more reproducible results.

If under these conditions IGX-MS does NOT lead to more robust results, requires less material or circumvents critical steps in sample productions, the authors should be careful to make such statements.

As stated before IGX-MS could still be very exciting and the reviewer appreciates the additional experiments on mitochondrial complex I by the authors.

However, it would be desirable if the authors could show that they are able to differentiate biologically relevant samples - and not only monomeric and assembly forms of a recombinant protein complex- by this method: for example, obtain data from different versions of a protein complex from a lysate.

Please find additional specific comments to the answers of the authors below:

In their manuscript "Selective cross-linking of coinciding protein assemblies by in-gel cross-linking mass-spectrometry" Hevler et al. investigate how cross-linking mass spectrometry (XL-MS) can be combined with blue native PAGE in-gel crosslinking (termed IGX-MS by the authors) in contrary to the conventionally used in-solution XL-MS.

The authors go on to demonstrate the applicability and use of IGX-MS on GroEL, ATP synthase from bovine heart mitochondria and the complement system. The authors show that IGX-MS, while overall leading yielding less cross-links that conventional in-solution XL-MS, leads to more robust results, circumvents critical steps in sample processing and, importantly, a substantial reduction in artificial or over-length cross-links.

While the idea of combining XL-MS with native PAGE separation is innovative and novel and highly intriguing, it is currently unclear if IGX-MS does indeed offer the advantages described by the authors and, more importantly, if those are indeed associated with potential novel biological insights that could not be gained from conventional XL-MS. Generally, it appears therefore doubtful if this approach, while exciting for the more MS-oriented community, could warrant a publication in EMBO J.

>We thank the reviewer for the critical evaluation of our work. We are pleased that the reviewer pointed out the novelty of our innovative approach. Although the reviewer does not appear to be thoroughly convinced about the value of our work, we believe that the additional data and text editions of the criticized parts provide clarification of the high significance of our work for a broad Embo Journal readership. We are convinced that our approach deserves close attention from the non-MS-oriented community. With this work, we also aim at open up cross-linking MS

technology specifically to biologists and biochemists who often have a minimal amount of sample, by far not sufficient enough to apply all desired structural analysis tools. Standard in-solution cross-linking MS experiments typically require hundreds to thousands of μg , which is, in many cases, the ultimate limitation. We demonstrate that IGX-MS can comfortably work with low μg amounts. Additionally, we show that IGX-MS can avoid generating undesired cross-links originating from artifactual protein assemblies or aggregates and provides more relevant specific structural information on selected protein complexes, from which new biological insights can be obtained.

Specifically

The authors convincingly show that IGX-MS leads to very reproducible crosslinks under varying crosslinker concentrations. However, in order to state that this is an advantage over in-solution XL-MS, the authors have to repeat these experiments with in-solution XL-MS and show that this is here not the case.

>The concentration dependency when cross-linking a protein sample in-solution is well-accepted and has been described in the literature. Most described protocols for in solution XL-MS include a cross-linker concentration optimization step to avoid reaction conditions that lead to over-cross-linking of the sample (Iacobucci et al., 2018; Klykov et al., 2018).

Regarding the argument for a simplified work-flow. This is a bit of a peculiar argument, as IGX-MS necessitates the use of an additional PAGE gel - such an extra, arguably quite time and work-intensive step. In contrary to in-solution XL-MS, where, as the name indicates, the sample is directly cross-linked and processed in solution.

>We respectfully disagree. The first step of in-solution cross-linking lies within finding an optimal cross-linker concentration (or a condition like pH or temperature - not discussed here), to check that cross-linking has occurred sufficiently but without over cross-linking the sample. Therefore, a subset of the sample is typically incubated with varying concentrations of cross-linker reagent. Cross-linked samples (and a non - cross-linked control) are subsequently analyzed by SDS-PAGE and potentially by BN-PAGE (especially useful for purified proteins). The aim is to determine a cross-linker concentration that produces cross-linked species without causing non-specific protein aggregation (usually visible as a smear on the gel or as thick band that has not entered the gel). It is essential to verify that no over cross-linking has occurred as cross-links from protein aggregates do not reflect the native protein conformations (Iacobucci et al., 2018; Klykov et al., 2018). Only after the optimal cross-linker concentration has been determined, the final in-solution cross-linking can be performed on a sufficient amount of sample. For IGX-MS, this sample and time-consuming optimization step is obsolete and concentration-independent in the tested concentration ranges. Thus, in-solution XL-MS requires at least two steps (concentration optimization and in-solution cross-linking experiment with optimal concentration), whereas IGX-MS only requires one step. We highlighted this problem in the main text with the following sentences: "Additionally, when using too high cross-linker concentrations or too high protein concentrations, undesired artificial interactions are likely being picked up by XL-MS. In-solution XL-MS experiments, therefore, need careful experimental optimization of, in particular, the concentration

of the proteins and the cross-linker. Unfortunately, these steps require considerable sample amounts (tens of micrograms) and extra experimental time." (page 2 - line 52-56)

>>The reviewer is aware of the cited literature. It is however misleading in this case. >>While it is certainly advantageous to optimize crosslinker concentrations it is also widely accepted that a ratio of crosslinker/ protein does exist that - at least in the case of recombinant proteins or protein complexes - leads to adequate results in nearly all of the cases tested (e.g. published) to date and usually does successfully avoid over crosslinking (1 to 2 ml/mg protein concentration at 1 to 2 mM crosslinker concentration).

>>Thus, the correct comparison would be to compare crosslinks from in-solution XL-MS under these standard conditions with IGX-MS - and then compare which workflow is simpler, requires less material and leads to more reproducible results.

Also, the other arguments, e.g. that there is no need for an extra buffer-exchange nor for time-consuming test of crosslinker concentrations are only half-convincing - see above - and as it has been shown that crosslinking also works in all kinds of buffer environments, including TRIS.

>We agree, that depending on the cross-linker's chemical structure, different buffer systems might work (including Tris). However, for the here applied reagents (DSS and DSSO), which reaction chemistry is amongst the most commonly used cross-linkers (NHS-reactive cross-linkers) (Steigenberger et al., 2020), it is crucial to avoid primary amines in the buffer system (e.g., present in Tris buffer). Such primary amines compete with the primary amines of the Lysine side chain and quench the reactive sites of the cross-linker, thereby significantly decreasing the reaction efficiency. Hence, it is crucial to remove any primary amine-containing reagent before a cross-linking experiment, meaning that if the desired sample is buffered with, e.g., Tris, an additional buffer exchange step is necessary to remove competing amines (Iacobucci et al., 2018; Klykov et al., 2018). As shown in the manuscript, this is not needed for IGX-MS, since the BN-PAGE serves as a natural buffer-exchange system (see, for instance, the GroEL result section). We further highlighted this in the main text: "Notably, the initial sample buffer (Tris) is incompatible with amine-reactive cross-linkers, as it contains primary amines that compete with the primary amines of Lysine residues, thereby significantly reducing the cross-link efficiency. Successful cross-linking of GroEL in-gel highlights the buffer exchange capacity of the BN-PAGE system, making additional buffer exchange steps (e.g., using molecular weight cut-off filters or dialysis), which would be required for in-solution cross-linking and eventually result in loss of sample, obsolete." (page 4 - line 103-109).

>>Again, the reviewer is aware of both the literature and the fact the NHS-reactive crosslinkers have been used. The reviewer would ask the authors to repeat their in-solution XL-MS experiments in TRIS under standard conditions with non-TRIS containing buffer system to find that crosslinking results will be comparable.

It is also not convincingly shown in the current manuscript, how IGX-MS can work with smaller amounts of sample than in-solution XL-MS - given the fact that it

requires this additional gel fractionation step.

>To illustrate the sample and time requirements to perform the cross-linking experiment, we describe in this context additional experiment on C6 incorporated in the revised manuscript as an example:

For IGX-MS, a total amount of 15 µg (5 µg/ replicate) of protein was used to generate a sufficient amount of distance restraints to drive structural modeling. For in-solution XL-MS of C6, we first optimized cross-linker concentration by incubating C6 with six different DSS concentrations (0-1.5 mM DSS), which resulted in 30 µg of C6 in total used for optimization (Figure EV3A). Next, 20 µg of C6 (in triplicates) was cross-linked with 0.25 mM of DSS in a total volume of 20 µL (ensuring comfortable sample handling and sufficient material for downstream processing such as digestion and de-salting), increasing the amount of used protein to 90 µg. Since cross-linking in-solution induced slight dimerization of C6, an additional SDS-PAGE separation was used to analyze monomeric, cross-linked C6, thereby further increasing the sample amount and experimental time.

>>See above: the correct comparison is to compare crosslinks from in-solution XL-MS under these standard conditions with IGX-MS - and then compare which workflow is simpler, requires less material and leads to more reproducible results.

The main argument of the authors, that IGX-MS leads to a substantial reduction in artificial or over-length cross-link, is indeed a potentially significant advantage, but also needs clarification. The authors compare their data on ATP synthase to an earlier dataset of their own (Liu et al., 2018). This is interesting, as the authors have explicitly stated in that paper that the majority of their crosslinks (137 of 139) was below the critical threshold of 30 Å. It would be nice if the authors were to clarify this point.

>We would like to clarify: The reported cross-links (139) referred to by the reviewer are derived from Complex I, Complex II, Complex III, and Complex IV as well as from Succinyl-CoA Synthetase and MAC-ETF complex - here a citation from Liu et al (Liu et al., 2018): "After interrogating the mitochondrial organization, we also assessed the structural validity of the cross-links based on the residue-to-residue distance limit imposed by the DSSO cross-linker. For this purpose, we selected six stable protein complexes (Succinyl-CoA synthetase, MCAD-electron transferring flavoprotein complex, and ETC CI, CII, CIII, and CIV) and mapped the identified cross-links onto available high-resolution structures."). Cross-link distances for the ATPase (PDB ID: 5ARA) presented in our study were not reported in the original manuscript by Liu et al., and therefore our data comparison with Liu et al. data set is relevant and correct. We specified the data set which we used from Liu et al. 2018 as follows: "The detected cross-links were plotted onto the 3D structure (PDB ID: 5ARA) and compared to cross-links detected in a previously published data set from our lab, by in-solution XL-MS (Liu et al., 2018) (Fig. 4B, SI Table S3 - contains data from Liu et al. 2018 used for the comparison)."

>>Apologies if there was a misunderstanding and for asking again: What is now the outcome of this new comparison? Does IGX-MS result now in a substantial reduction in artificial over-length cross-links or not?

For the other datasets generated in this study - e.g. on GroEL - would the amount of over-length crosslinks for in-solution XL-MS be reduced if the authors were to crosslink GroEL at lower concentration? Or in other words - the authors have used relatively large amounts of sample at relatively high-concentrations - and according to the workflow the sample concentrations during in-solution XL-MS (prior to native PAGE) were obviously different to the ones used for IGX-MS (e.g. after native PAGE). This should be taken into account.

>We believe, if lower cross-linker concentrations were to be used for in-solution XL-MS the general number of identified cross-links would most likely go down (including both, cross-links satisfying- and violating the distance restraint). Also, optimization of the DSS concentration for GroEL (Appendix Fig 3A) in-solution cross-linking experiments did not indicate protein aggregation, which would subsequently result in a higher proportion of cross-links that are disagreeing with the structural model. The concentration of GroEL during in-solution XL-MS was chosen to be 1 mg/ml. This is the concentration that was also used for the optimization of the cross-link reaction (Appendix Fig 3A), producing different species of cross-linked GroEL subunits, without visible aggregate formation (Appendix Fig 3A). For IGX-MS 10 ug GroEL was loaded on the BN-PAGE. The concentration of GroEL during the cross-linking in the gel is determined by the size of the gel band, which was not measured. We can therefore not comment on the comparison on the concentrations as mentioned by the reviewer. However, we are convinced that we designed the experiment rigorously. We carefully evaluated the cross-linker concentration's influence on the successful cross-linking reaction in IGX-MS (Fig 2). Based on these results, we can conclude that IGX-MS provides a high number of relevant cross-links and substantially decreases the amount of over-length cross-links with no need to optimize the cross-linker concentration.

>>This is indeed not a major point - even though it should be possible to estimate the effective protein concentrations within the native gel and then carry out in-solution XL-MS under these conditions for comparison.

Still, IGX-MS, also in the eyes of the reviewer, is a truly neat idea and could hold the potential to generate biological meaningful data. However, in order to prove this, the authors should carry out an additional experiment showing that is possible to obtain data with IGX-MS that in-solution XL-MS is not able to generate. For example, perform IGX-MS of a cell lysate and demonstrate that IGX-MS can differentiate between different populations of the same complex.

>To further showcase the ability of IGX-MS to distinguish between different populations of the same complex, we additionally applied IGX-MS on two assembly states of the mitochondrial complex I (Figure EV1). Like for the ATPase, CI protein populations were separated by BN-PAGE and bands corresponding to the monomeric complex I and the complex I in an assembled "super-complex" state with complex III and complex IV (S1 complex - I1-III2-IV1) were isolated and subjected to IGX-MS (Figure EV1A). IGX-MS enabled us to distinguish between cross-links that were only detected in either one of these assembly states (monomeric complex 1 or in S1) and cross-links that were detected for both assembly states (Figure EV1A-B). Again, IGX-MS generated data showing an agreeable overlap with previously generated in-solution data (Liu et al., 2018), with the advantage of distinguishing the

protein (super) complex the cross-link was detected in (Figure EV1B). We hope this data may help to convince the reviewer that IGX-MS has indeed, in several exciting cases, advantages over in solution XL-MS.

>>The reviewer appreciates these additional experiments by the authors.

>>Still, in order to show the relevance of IGX-MS, the author should apply IGX-MS to biologically more relevant samples (for example, obtain data from different versions of a protein complex from a lysate).

Thanks once more for handling our manuscript and providing feedback. I discussed it now also further with the co-authors. We are delighted to note that the reviewers acknowledge the novelty and relevance of the IGX-MS methodology described in our manuscript. We also appreciated all the reviewer's remarks and comments, which substantially have improved the manuscript. I am afraid that my response is a bit longer, but we must also say that we were a little annoyed and upset by the final comments of the reviewer.

In the following paragraphs, we will provide comments on the persisting concerns of reviewer 3, especially regarding a direct comparison between IGX-MS and in-solution XL-MS. As discussed before with you we did expect this possibly to happen, but are also pleased to see that reviewer 2 does not support the persistent issue of reviewer 3.

First, we want to stress (once more, as discussed with you before) that a direct comparison of the two cross-linking approaches has not been major driver in our study. IGX-MS was not designed to replace – or to be an all-in-all superior methodology to in-solution XL-MS. We developed with IGX-MS a convenient approach to target specific protein assemblies when multiple protein assemblies are co-occurring in a sample, especially when only limited sample amounts are available. In the manuscript, we primarily performed in-solution XL-MS to validate the cross-links obtained with IGX-MS and highlight certain advantages and benefits of applying IGX-MS. We showcase that in the analyzed biological systems (e.g., complex V of bovine heart mitochondria), IGX-MS can specifically target protein assemblies and generate distance restraints consistent with in-solution XL-MS. Next, we detected cross-links that well agreed with available protein structures (e.g., monomeric complex V - PDBID: 5ARA) and compared to in-solution XL-MS, we observed a significantly lower number of over-length cross-links. We conclude that the benefit of reduced over-length cross-links lies within the ability to target a specific assembly state (e.g., monomeric complex V), which with XL-MS would not be possible without further experimental fractionation (as stated in the manuscript). As the manuscript aims not to down play the great features of in-solution XL-MS but rather to highlight the possibility and benefits of generating assembly specific cross-links in a gel, we do not see additional improvement of our manuscript by conducting a "fair" comparison according to the reviewer's proposal. We also would like to point out that performed in-solution XL-MS experiments (including cross-linker optimization) are in line with published "best practice" protocols. We are not aware of any literature or protocols describing "widely accepted cross-linker/protein ratios." To support this, we also want to refer here to the latest cross-linking meeting report "Toward Increased Reliability, Transparency, and Accessibility in Cross-linking Mass Spectrometry" (<https://doi.org/10.1016/j.str.2020.09.011>) that recommends control experiments to address oligomerization or sample integrity (hence making a cross-linker optimization an essential step of in-solution XL-MS). Therefore, we and most of the community is convinced that optimization of reaction conditions is an integral part of the in-solution XL-MS protocol and needs to be included for potential comparing in-solution XL-MS and in-gel IGX-MS. From that point of view, our statements regarding the advantages of IGX-MS, such as lower sample consumption or fewer optimization steps, are justifiable and relevant.

The reviewer also commented on our data interpretation from the manuscript Liu et al., 2018. However, this was not relevant due to his/her mistaken reference to the data. Thus, the data used for the comparison did not change during the revision of the manuscript. In the manuscript, we clarified which dataset was used and provided an exhausting interpretation of obtained results with comprehensive data visualization in the main text and the supplementary material.

Lastly, reviewer 3 is not convinced that the samples analyzed in our manuscript were not biologically relevant enough. We want to stress that mitochondrial and complement proteins are essential proteins involved in critical biological processes, and their deeper understanding is crucial. The mitochondrial sample is a complex system consisting of several proteins and is ideal for demonstrating IGX-MS capabilities to handle complex samples. We not only showcase the ability to cross-link assembly specific states of mitochondrial proteins in gel, but also provide an alternative structural model of complement C6, which is additionally supported by previously published data. Hence we believe that further analysis of lysates or other samples is beyond the scope of this work. Additionally, we are convinced that an experiment tackling Tris buffer's incompatibility with DSS and DSSO does not provide relevant improvement of the manuscript as anyone should realize that buffers containing free amines are a no-go with NGS-esters.

Conclusively, we are not aware of making any statement regarding the superiority of IGX-MS over in-solution XL-MS in general in our manuscript, as it is not in the scope of this work. We are presenting a methodology that showcases a possibility to circumvent some of the challenges for in-solution XL-MS. Of course, we are also aware of some of the caveats of IGX-MS (as mentioned in the manuscript). Nevertheless, we value and understand the reviewer's 3 comments and are willing if you insist to further clarify the highlighted purpose of IGX-MS in the manuscript, as a neat methodology to generate useful assembly specific distance restraints and not as a superior- nor replacement method for in-solution XL-MS.

This would then mean that we include the highlighted sentence below in the discussion.

The work presented here collectively describes a novel methodology termed IGX-MS, which allows the efficient, sensitive, and reproducible generation of specific structural distance restraints. The methodology described here should not be regarded as replacement for in-solution XL-MS, but rather as a convenient, alternative approach. IGX-MS can best be used on proteins and protein complexes that can be well-separated in a BN-PAGE gel and this is not the case for all proteins. Nevertheless, when amendable to BN-PAGE, IGX-MS provides the ability to distinctively analyze co-occurring protein oligomers in purified systems, even when originating from more complex samples, such as solubilized mitochondria.

We find we have addressed the concerns of reviewer 3, and feel supported by reviewer 2, and hopefully now also by you as editor. We are looking forward to hearing back from you.

2nd Editorial Decision**27th Nov 2020**

Thank you for your detailed and well-laid out response to the second round of reviewer comments. I have now considered them, also in light of the additional feedback we received from referee 2, and decided to accept the manuscript following incorporation of the additional discussion sentence as proposed in your letter. In addition, there are also a few remaining editorial points that should be incorporated during a final minor revision round.

2nd Authors' Response to Reviewers**3rd Dec 2020**

The authors have made all requested editorial changes.

Accepted**10th Dec 2020**

Thank you for submitting your final revised manuscript for our consideration. I am pleased to inform you that we have now accepted it for publication in The EMBO Journal.

Corresponding Author Name: Albert J.R. Heck

Journal Submitted to: Embo Journal

Manuscript Number: EMBOJ-2020-106174